# Unlocking Out-of-Distribution Generalization in Transformers via Latent Space Reasoning

## Abstract

Systematic, compositional generalization beyond the training distribution remains a core challenge in machine learning—and a critical bottleneck for the emergent reasoning abilities of modern language models. This work investigates out-of-distribution (OOD) generalization in Transformer networks using a GSM8K-style modular arithmetic circuit-evaluation task as a testbed. We introduce and explore a set of four architectural mechanisms aimed at enhancing OOD algorithmic generalization: *(i)* input-adaptive recurrence; *(ii)* algorithmic supervision; *(iii)* anchored latent representations via a discrete bottleneck; and *(iv)* an explicit error-correction mechanism. Collectively, these mechanisms yield an architectural approach for native and scalable latent space reasoning in Transformer networks with robust algorithmic generalization capabilities. We complement these results with mechanistic interpretability analysis showing how these mechanisms yield robust OOD generalization.

## 1 Introduction

Systematic algorithmic generalization stands as a critical milestone and a grand challenge in machine learning research [Pol90; Soc+12; LB18; VB21]. This ability is fundamental to human cognition, stemming from our capacity for *systematic compositionality*—algebraically producing novel combinations from known components and making strong generalizations from limited data [Cho57; FP88; Lak+17]. Achieving such generalization necessitates learning universal, scalable problem-solving algorithms. Even in humans, acquiring such algorithmic understanding often requires explicit step-by-step supervision. Once an algorithm is learned, however, humans can generalize its application far beyond previously encountered domains [And82; SA89].

The reasoning capabilities of artificial intelligence systems have advanced rapidly in recent years, built upon the foundation of large language models. In particular, chain-of-thought (CoT) techniques have been central to enhancing these capabilities [Wei+22; Koj+22; Chu+22; Liu+23], especially in domains like mathematics [Cob+21; Lew+22; Lig+23; Sha+24]. CoT provides supervision for learning a reference problem-solving procedure during training and allows the model to emulate it at test time. This progress presents an opportunity to make significant strides on foundational challenges in artificial intelligence reasoning.

However, despite these advances, out-of-distribution (OOD) generalization—particularly the type of *length generalization* involved in algorithmic reasoning (i.e., generalizing from simpler or smaller problem instances to larger or more complex ones)—remains a central challenge for Transformer-based [Vas+17] language models [Ani+22; Kaz+23; Jel+23; Zho+24a; Tho+24; Dzi+23]. While chain-of-thought techniques alleviate this to some degree by enabling the learning of more complex algorithmic procedures, the ability to generalize far outside the training distribution remains a significant obstacle [SVK24; Zho+24b].

In this work, we investigate the architectural and methodological mechanisms that underpin algorithmic OOD generalization in Transformer networks. To facilitate a systematic investigation, we focus our study on a simple yet scalable mathematical reasoning task: evaluating modular arithmetic circuits. This task allows us to study OOD and algorithmic generalization in a controlled manner, with complexity directly parameterized by circuit size and depth, while also capturing the essence of established mathematical reasoning benchmarks like GSM8K [Cob+21] that are central to evaluating the reasoning capabilities of large language models. Furthermore, this task possesses a compositional nature; it can be solved by learning a core set of skills and

scaling up their application to solve larger and more complex problem instances. We use this task to explore the following guiding question: *What are the architectural mechanisms and inductive biases needed for robust and systematic OOD algorithmic generalization in Transformers?*

We find that while standard CoT training techniques enable good in-distribution performance and a limited degree of OOD generalization, the learned solutions are not robust or universal, and performance rapidly degrades as test inputs grow in complexity beyond the training regime. We propose and explore a set of four simple architectural and methodological mechanisms, built upon the Transformer architecture, to facilitate the learning of robust and generalizable algorithmic solutions: *(i)* input-adaptive recurrence; *(ii)* algorithmic supervision; *(iii)* anchored latent representations via a discrete bottleneck; and *(iv)* an explicit error-correction mechanism. When combined, these mechanisms yield an architectural approach for native and scalable *latent space reasoning* in Transformer networks, demonstrating robust algorithmic generalization capabilities. In particular, on our mathematical reasoning task, our method achieves perfect generalization on inputs that are several times larger than those seen during training. We complement our architectural proposal and empirical results with a mechanistic interpretability analysis to reveal *how* these mechanisms enable sharp OOD generalization, what circuits they learn, and why those circuits facilitate robust OOD generalization.

## 2 Related Work

Our work is related to several strands of fundamental machine learning research, including issues of out-of-distribution generalization, architectural mechanisms such as recurrence and discretization, chain-of-thought and intermediate supervision methods, and work on mechanistic interpretability techniques.

**Out-of-Distribution Generalization.** Out-of-distribution (OOD) generalization, along with related capabilities such as compositionality and systematicity, poses a fundamental challenge in machine learning research [Pol90; Bax00; Soc+12; Bar+18; Hup+20]. These capabilities are crucial for developing AI systems that can reliably apply learned knowledge to novel scenarios, a hallmark of robust intelligence [FP88; Lak+17; GB22]. A particularly important type of OOD generalization, especially for algorithmic reasoning tasks, is *length generalization*—the ability to generalize from simpler or shorter training instances to significantly longer and more structurally complex instances. This has proven to be a key limitation of Transformer-based [Vas+17] language models [Ani+22; Kaz+23; Jel+23; Zho+24a]. While chain-of-thought techniques alleviate this to some degree by enabling the learning of more complex algorithmic procedures, the ability to generalize far outside the training distribution remains a significant obstacle [SVK24; Zho+24b].

**Recurrence.** Recurrence forms a foundational architectural principle in neural networks, particularly for tasks that involve sequential data or inherently iterative processes [Elm90; Jor97; HS97]. These architectures are designed to emulate step-by-step computations by maintaining and updating an internal state, making them well-aligned with problems that have a recursive or layered solution structure. Sequence-to-sequence recurrent architectures for sequence transduction and neural machine translation advanced the state of the art [Cho+14; SVL14], and were instrumental to the development of attention mechanisms and the Transformer architecture [Vas+17]. While standard Transformers do not possess a recurrent structure, recurrent variants of the Transformer architecture were explored soon after its introduction [Deh+19]. Whereas standard recurrent neural networks apply their recurrence across time or sequence length, recurrent Transformer architectures are *parallel in time* due to the parallel attention mechanism, but recurrent across computational depth—that is, the same Transformer layer is applied iteratively to the sequence as a whole. The recurrent inductive biases have been demonstrated to confer certain advantages in generalization [Fan+24; Yan+24; Wan+25; Jol25]. In our work, recurrence is a key architectural mechanism encoding important inductive biases that aid the discovery of scalable recursive algorithms for solving the underlying mathematical problem.

**Adaptive Computation.** A critical challenge is handling inputs with varying complexity, where a fixed amount of computation may be inefficient or insufficient. This motivates *adaptive computation*, where a model dynamically adapts its computation time based on input demands, for example by varying recurrent iterations. An important work in this domain is the *Adaptive Computation Time (ACT)* mechanism proposed by Graves [Gra17] for recurrent neural networks, which explicitly models and learns how many computational steps are needed as a function of the input. A version of the ACT mechanism is incorporated in the recurrent Transformer architecture proposed by Dehghani et al. [Deh+19]. However, a drawback of such mechanisms

is their complexity and difficulty of training. Although efforts have been made to explore simpler adaptive computation methods [BBB21], an even simpler approach is explored by Schwarzschild et al. [Sch+21] and Bansal et al. [Ban+22], where the halting time is not explicitly modeled by the network, and instead the number of recurrent iterations is scaled at inference time based on the size of the input. This simpler approach can be easier to train, and has been shown to improve out-of-distribution generalization. More recently, Geiping et al. [Gei+25] explored the viability of this approach for test-time scaling in large language models. In our work, we similarly scale computation time by proportionately scaling the number of recurrent iterations in order to solve more complex problem instances, generalizing far beyond the training distribution.

**Discreteness in Neural Networks.** Symbolic AI systems derive their power from manipulating discrete symbols according to well-defined rules, which enables robust, precise, and interpretable reasoning [NS76; FP88]. Given this rich tradition of discrete symbolic states in artificial intelligence, many works have explored incorporating such discrete latent representations into neural networks [GLG08; SH09; CBB11; Agu+17; OVK18]. Discreteness is also a key characteristic of *constructions* of Transformers for specific tasks. For example, Weiss, Goldberg, and Yahav [WGY21] develop a programming language that represents Transformer-based computation with discrete internal mechanisms, while Smolensky et al. [Smo+24] construct a Transformer network for a compositional in-context learning task with discreteness in both its latent states and attention mechanism. In our work, we explore the use of discrete latent states as a means of *anchoring* the latent representation to a common, depth-invariant space to enable scaling computation far beyond the training distribution while avoiding representational shift across computational depth.

**Chain-of-Thought & Algorithmic Supervision.** Chain-of-thought techniques have been central to enhancing the reasoning capabilities of large language models. Early usage of the term "chain-of-thought" referred to prompting techniques that condition a model to generate a sequence of intermediate steps before arriving at the final answer [Koj+22; Wei+22; Nye+21]. For example, Wei et al. [Wei+22] demonstrated that prompting the LLM with a few CoT exemplars caused the model to generate an analogous step-by-step solution, which significantly improved performance on a range of arithmetic, commonsense, and symbolic reasoning tasks. Kojima et al. [Koj+22] showed that LLMs can be "zero-shot" reasoners in the sense that simply asking the model to reason step-by-step, without providing in-context learning CoT exemplars, can be sufficient to elicit chain-of-thought-style reasoning and improve performance. Modern usage of the term "chain-of-thought" has extended beyond prompting methods, as it now forms a key component of the *training* pipeline of LLMs, wherein a model is explicitly trained on demonstrations of step-by-step solutions to problems of interest, such as mathematical reasoning [Chu+24; Liu+23; Lew+22]. In some situations, chain-of-thought training can be interpreted as providing explicit supervision to align the model to a particular algorithm or procedure for solving a problem, as opposed to simply providing supervision via input-output examples. In our work, we explore traditional chain-of-thought training techniques as baselines, as well as incorporate algorithmic supervision to the internal states of our proposed method.

**Mechanistic Interpretability.** In our work, we carry out a mechanistic interpretability analysis to probe *how* the model has learned to solve the task and *why* it can do so robustly, generalizing far outside the training distribution. In recent years, there has been a resurgence in work on interpretability, with new techniques being introduced that aim to understand modern large language models [Elh+21; Men+22; Elh+22; Ols+22; Bri+23; Ame+25]. Elhage et al. [Elh+21] is an influential work in this area of research, introducing a conceptual framework and new terminology that continues to be used in subsequent work. A key early achievement in this line of work is the discovery of "induction head" circuits in large language models [Ols+22], which perform a two-step copying operation that is crucial for in-context learning. In our work, we identify a similar mechanism in our recurrent models that is used to copy previously computed variable values. This involves first retrieving the parent variables' names in the first layer, then using these variable names to retrieve their values in the second layer, which are computed elsewhere in the sequence of latent states. Such work is often described as *circuit analysis*, where the goal is to identify sub-networks that are responsible for particular functions. A key method for validating hypotheses about the functions of different model components is *causal interventions* like activation patching or ablations [Men+22; Gei+21; Gei+24], which involves systematically modifying parts of the model or input to observe effects on behavior or internal states. We use related causal intervention techniques in our own mechanistic interpretability analysis in this work. Finally, the work by Nanda et al. [Nan+23] and Tian [Tia24] is relevant as it specifically

investigated how Transformers perform arithmetic, reverse-engineering a modular addition algorithm learned by the feedforward network in a Transformer layer, a phenomenon we also observe in our models.

# 3 Problem Setup

## 3.1 Task Description: Modular Arithmetic Circuits

We formally introduce our *modular arithmetic circuits* task as follows.

**Task Description.** A problem instance is a *modular arithmetic circuit*: a directed acyclic computation graph (DAG) that specifies a network of modular arithmetic computations. Nodes correspond to variables, and edges encode the dependencies between them. As illustrated in Figure 1, *leaf nodes* are directly assigned values (e.g., $x_7 \leftarrow 20$). All other *non-leaf nodes* are defined as functions of their parent nodes. In particular, each non-leaf node is computed by applying specified operations to the values of its parent nodes. In our experiments, we consider *modular arithmetic* operations (addition, multiplication, or subtraction) with prime modulus $p = 23$. For example, in Figure 1 we have $x_{23} \leftarrow x_7 + x_{42} \pmod{p}$ and $x_{101} \leftarrow x_{23} \times x_{91} \pmod{p}$. In what follows, let $N$ be the total number of nodes and $L$ the number of leaf nodes. We consider circuits with up to $N = 128$ nodes and use $\mathcal{V} = \{x_1, \ldots, x_{128}\}$ to denote the set of variable names.

**Data Generation Process.** A problem instance in this task is specified by the *values of the leaf nodes* and *a circuit* depicting the computations that determine the values of all non-leaf nodes. In particular, given parameters $N$ and $L$, an input instance is generated as follows:

  (i) Randomly generate a DAG with $N$ nodes, $L$ of which are leaf nodes.
 (ii) Randomly assign a variable name from $\mathcal{V}$ to each node.
(iii) Randomly assign numerical values to the leaf nodes from $\mathcal{N} = \{0, 1, \ldots, 22\}$.
(iv) For each non-leaf node, randomly assign operations from $\mathcal{O} = \{+, -, \times\}$ to define its computation based on its parent nodes

The instance generated by (i)–(iv) is stored as a token sequence, where each variable name, numerical value, and operation is assigned a unique token. A special *separation token* [sep] is used to separate different formulas. For example, the instance depicted in Figure 1 is represented as the following *token sequence*:

$$
\begin{aligned}
&\langle 20 \rangle \langle \rightarrow \rangle \langle x_7 \rangle \,[\texttt{sep}]\, \langle 2 \rangle \langle \rightarrow \rangle \langle x_{42} \rangle \,[\texttt{sep}]\, \langle 6 \rangle \langle \rightarrow \rangle \langle x_{88} \rangle \,[\texttt{sep}]\, \langle 14 \rangle \langle \rightarrow \rangle \langle x_{115} \rangle \\
&\langle x_7 \rangle \langle + \rangle \langle x_{42} \rangle \langle \rightarrow \rangle \langle x_{23} \rangle \,[\texttt{sep}]\, \langle x_{42} \rangle \langle + \rangle \langle x_{88} \rangle \langle \rightarrow \rangle \langle x_{91} \rangle \,[\texttt{sep}]\, \langle x_{88} \rangle \langle \times \rangle \langle x_{115} \rangle \langle \rightarrow \rangle \langle x_{55} \rangle \,[\texttt{sep}]\, \\
&\langle x_{23} \rangle \langle \times \rangle \langle x_{91} \rangle \langle \rightarrow \rangle \langle x_{101} \rangle \,[\texttt{sep}]\, \langle x_{91} \rangle \langle - \rangle \langle x_{88} \rangle \langle + \rangle \langle x_{55} \rangle \langle \rightarrow \rangle \langle x_{30} \rangle
\end{aligned}
\tag{1}
$$

**Target Output & Evaluation Metric.** Given a problem instance, the task is to compute the value of every node in the circuit; these values are uniquely determined by steps (i)–(iv) above. We consider the model output to be correct only if *all* node values are computed correctly (i.e., the input circuit is fully solved).

**Out-of-Distribution Generalization.** Our primary focus in this work is to investigate the ability of Transformer networks to learn general procedures or algorithms that enable *out-of-distribution* (OOD) generalization. The complexity of each problem instance can be explicitly parameterized by circuit size, enabling precise measurement of a model's ability to generalize to inputs more complex than those encountered during training. In particular, in this task, OOD generalization is evaluated by training models on problem instances with $N \leq 32$ nodes and testing them on instances of varying sizes, up to $N = 128$ (a fourfold increase). Such generalization requires the ability to process larger inputs and adaptively scale computation time during testing, beyond what was encountered in the training regime.

**Representativeness and Generality of the Modular-Arithmetic Circuit Task.** In addition to providing fine-grained control over instance complexity, this circuit formulation captures structural properties shared by a broad class of algorithmic reasoning tasks. Any program over a finite primitive operation set can be represented as a directed acyclic computation graph, or circuit, in which nodes encode intermediate computations and edges specify data dependencies. Depth in such circuits corresponds to the number of

sequential computational steps required to solve an instance, making depth extrapolation a direct proxy for the central challenge in algorithmic reasoning: maintaining stable computation as compositional depth grows. Although our instantiation uses modular arithmetic as the primitive operations, the underlying abstraction—symbolic computation over DAGs—is general. Moreover, this synthetic task retains the essential combinatorial structure of mathematical-reasoning benchmarks such as GSM8K [Cob+21], while eliminating incidental complexities of natural language and enabling precise analysis of the learned computation (Section 5).

## 3.2 Limitations of Standard Transformers with CoT Training

To establish baselines and motivate the need for alternative approaches, we evaluate standard Transformer architectures on our synthetic task under two training paradigms.

**End-to-End Training.** The first baseline is *End-to-End* training, where models are trained to directly output the final values of all nodes from the input, without explicit intermediate steps. We consider feedforward (fixed depth) and recurrent (fixed recurrence) architectures.

**Chain-of-Thought (CoT) Training.** The second baseline is autoregressive *Chain-of-Thought (CoT)* training [Wei+22; Cob+21; Lew+22; Chu+24; Ye+24], a prevalent technique for enabling multi-step reasoning in LLMs. Instead of directly outputting the final answer, CoT trains a model to generate a sequence of intermediate reasoning steps (the "thought process") that culminates in the solution. For our task, CoT intermediate steps explicitly demonstrate the step-by-step evaluation of nodes in a given circuit. In particular, under *CoT* training, the Transformer model receives an input prompt consisting of

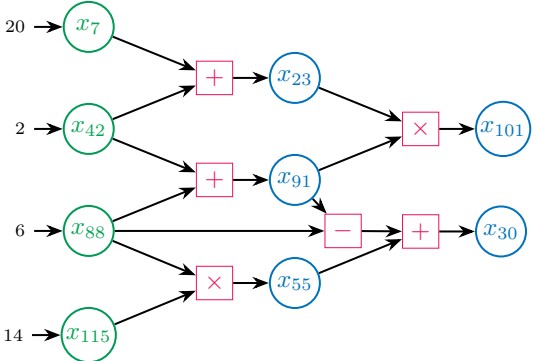

Figure 1: An illustration of a modular arithmetic circuit instance. The goal is to compute the values of all nodes in the circuit. For example, here, $x_{23} = 20 + 2 = 22$ and $x_{55} = 6 \times 14 = 15$.

the token representation of the circuit, followed by a special $\langle \text{CoT} \rangle$ token that signals the beginning of the CoT reasoning. The CoT then outlines the computation of each node in topological order. Each step in the trajectory involves: (1) recalling the equation defining the node's value, (2) recalling the values of its dependent (parent) nodes, and (3) performing the arithmetic computation. For example, computing node $\langle x_{101} \rangle$ from Figure 1 would appear in the CoT as:

$$[\ldots \texttt{Input Prompt}\ldots]\langle\texttt{CoT}\rangle[\ldots]\langle x_{101}\rangle = \langle x_{23}\rangle\,\langle\times\rangle\,\langle x_{91}\rangle = \langle 22\rangle\,\langle\times\rangle\,\langle 8\rangle = \langle 15\rangle$$

Here, $[\ldots \texttt{Input Prompt}\ldots]$ gives the description of the problem instance, and $[\ldots]$ denotes the preceding portion of the chain-of-thought trajectory up to node $\langle x_{91}\rangle$ (including the computation of $\langle x_{23}\rangle$ and $\langle x_{91}\rangle$). An example of a full CoT trajectory from the training data is provided in Appendix A.2.

**Implementation.** We train causal Transformer models from scratch using both *End-to-End* and *CoT* supervision on randomly generated instances with circuit sizes $N \leq 32$. At inference time, models are prompted with the input and generation is performed using *greedy decoding*. End-to-End models directly output all node values, while CoT models autoregressively generate the solution, including the full CoT trajectory. We evaluate performance by the proportion of instances for which the model computes *all* node values correctly, focusing on OOD generalization to new, randomly generated circuits of varying sizes up to $N = 128$. For each method, we perform an extensive hyperparameter search (including layers, model dimension, and positional encoding) to ensure that we compare against the best-achievable performance for each method. For example, prior work observed that positional encoding methods and other hyperparameter choices can have a significant impact on out-of-distribution generalization performance [CIS21; Kaz+23]. Experimental details for these baselines are provided in Appendix A.

**Observed OOD Generalization Deficiencies.** We find that *Chain-of-Thought* training enables models to solve larger circuits compared to those trained *End-to-End* without chain-of-thought supervision (Figure 3a). However, even the best CoT models exhibit only *limited OOD generalization*: when trained on circuits with $N \leq 32$, performance extends only to moderately larger circuits (roughly $N \approx 40$), and then deteriorates

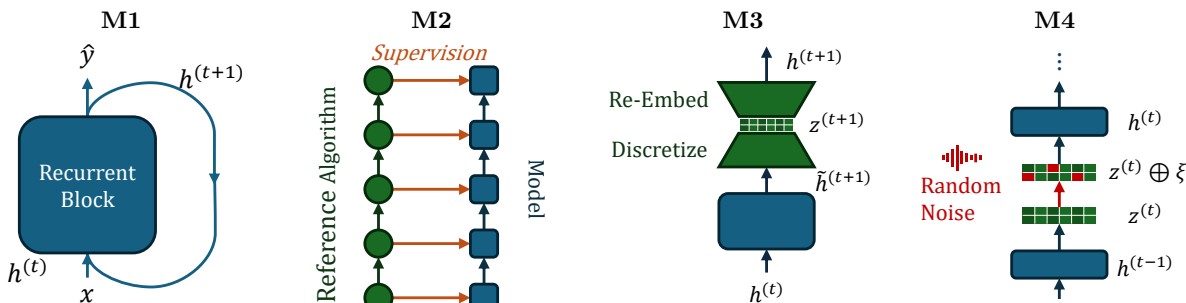

Figure 2: Depiction of four mechanisms for out-of-distribution generalization: **M1** recurrence and adaptive computation, **M2** algorithmic supervision, **M3** anchored discrete latent space, and **M4** error correction.

rapidly as $N$ increases beyond the training regime. In the next section, we propose a series of architectural mechanisms that address these generalization challenges.

**Theoretical view on the limitations of fixed-depth transformers on this task.** We note that our task of evaluating modular arithmetic circuits is related to evaluating Boolean circuits. In particular, logical gates can be expressed as arithmetic operations mod $p = 2$ (e.g., $\texttt{AND}(x, y) = x \cdot y$). Evaluating Boolean circuits is P-complete under log-space reductions [Lad75]. This suggests that no polylog-depth parallel algorithm exists for general modular-arithmetic circuit evaluation unless $P = NC$. Moreover, recent work has shown that constant-depth transformers (with logarithmic precision; without CoT) are contained in the class $TC0$, which is believed to be strictly smaller than $P$. This provides a theoretical view on why we need to go beyond traditional constant-depth Transformers to solve the modular arithmetic circuits task.

## 4  Reasoning in Latent Space with Algorithmic Supervision

### 4.1  Mechanisms for Effective OOD Generalization.

Effective OOD generalization on reasoning tasks hinges on a model learning a scalable *algorithm*: an iterative procedure that adapts to input complexity. Designing *inductive biases* that support the discovery of such scalable, compositional solutions is a central challenge in machine learning [Bax00; LB18; Bar+18; GB22]. Chain-of-thought (CoT) techniques address this by training the model to generate a token-level trace of a computation. However, forcing computation into an autoregressive token format often yields brittle "algorithms" that fail to generalize robustly, especially when longer CoT sequences are needed for more complex inputs. These length generalization issues [Ani+22; Kaz+23; Jel+23; Zho+24a; SVK24; Zho+24b] underscore CoT's limitations in emulating scalable algorithmic procedures. We therefore propose alternative mechanisms to facilitate learning such iterative algorithms directly within a model's latent dynamics.

Our proposal features recurrent Transformer blocks, algorithmic supervision, latent-space discretization, and a self-correction scheme (Figure 2). Together, these mechanisms yield an architecture with native latent-space reasoning and effective OOD generalization. In the following, we present the four mechanisms and the essence of their implementation, deferring certain details to Appendix B.

**Algorithm to Emulate.** To solve this task, a natural algorithmic solution—well-aligned with the Transformer architecture—is to compute the values in the circuit one layer at a time. This can be realized through a recursive process that iteratively applies the same computational module. Specifically, each iteration computes values one layer deeper in the circuit by fetching dependent values for nodes at the current layer and then performing modular arithmetic. In particular, for the example in Figure 1, in the first iteration we evaluate $\{x_7, x_{42}, x_{88}, x_{115}\}$; in the second iteration we evaluate $\{x_{23}, x_{91}, x_{55}\}$; and in the final iteration we evaluate $\{x_{101}, x_{30}\}$. Each iteration repeats this computation pattern, yielding a scalable algorithm.

**Mechanism 1: Recurrence & Input-Adaptive Computation.** The target layer-by-layer algorithm is iterative, naturally motivating a recurrent architecture. We employ a recurrent Transformer block so that each application emulates one algorithmic iteration—that is, computing values for one additional layer of the circuit. We represent an input instance as a length-$n$ token sequence $X = (x_1, \dots, x_n)$ (as described in Section 3),

embed it as $E_1^{(0)}, \ldots, E_n^{(0)}$, and apply a recurrent Transformer block RecurrentTransformerBlock:

$$(E_1^{(t+1)}, \ldots, E_n^{(t+1)}) \leftarrow \text{RecurrentTransformerBlock}(E_1^{(t)}, \ldots, E_n^{(t)}), \qquad (2)$$

for iterations $t = 0, 1, \ldots, T{-}1$. We linearly read out the output from the final embedding states $E_1^{(T)}, \ldots, E_n^{(T)}$. Crucially, the number of recurrent iterations, $T = T(X)$, is not fixed but adapts to input complexity, scaling linearly with the depth of the circuit. This enables computation time to grow with problem size, a key capability for OOD generalization to larger circuits. Unlike CoT methods that scale computation by generating progressively longer token sequences, recurrence reuses the same module over time and introduces inductive biases that favor scalable recursive solution structures. The use of recurrence to adaptively scale computation time for tasks with variable complexity is a recurring theme in the literature [Sch12; Ben+15; Gra17; Deh+19; BBB21; Sch+21; Ban+22; Fan+24; Gei+25; Wan+25; Jol25].

**Mechanism 2: Latent State Algorithmic Supervision.** While recurrence provides the capacity for iterative computation, it does not inherently guarantee that the model will learn a scalable layer-by-layer algorithmic procedure. To instill this structure, we introduce latent state algorithmic supervision. Unlike CoT, which supervises intermediate computation in token space, our approach provides supervision directly in the model's latent space at each recurrent step, steering the internal states to follow the step-by-step execution of our target algorithm. Specifically, at each recurrent iteration $t$, we use a shared *linear readout layer* to predict node values from the latent embeddings $E_i^{(t)}$ and apply a step-wise loss that aligns the model with the target layer-by-layer algorithm:

$$\text{AlgorithmAlignmentLoss} = \sum_{t=1}^{T} \sum_{\substack{i \in [n] \\ \text{Depth}(x_i) \leq t}} \ell\left(W_{\texttt{value}}\, E_i^{(t)}, \text{Value}(x_i)\right). \qquad (3)$$

where $\text{Depth}(x_i)$ is the node depth in the circuit, $\text{Value}(x_i)$ is its ground-truth value, $W_{\texttt{value}}$ is the shared readout, and $\ell$ is the cross-entropy loss. Thus, the algorithm alignment loss supervises the model such that at iteration $t$, it computes the values of all nodes in the input at computational depth less than or equal to $t$. Viewed differently, this is *depth-indexed* intermediate supervision: rather than supervising a token-level trace as in CoT, we supervise the internal state according to the algorithm's layer-by-layer computation schedule. For example, in Figure 1, supervision at $t = 1$ applies to leaf nodes (e.g., $x_7$), while at $t = 2$ it extends to include second-layer nodes (e.g., $x_{23}$), and so on. This iterative supervision encourages the model to progressively build up the solution, computing the circuit one layer deeper with each recurrent step.

Beyond the end-to-end label, implementing the algorithm alignment loss above requires only each node's depth in the circuit, specifying the order in which nodes ought to be computed. This is a weaker supervision oracle than the CoT baseline, which requires token-level traces that break down each variable's computation into (i) choosing its parent variables, (ii) retrieving their values, and (iii) applying the arithmetic operation.

**Mechanism 3: Anchoring Latent Representation via Discretization.** Recurrent models can suffer representational drift over long OOD rollouts, as errors accumulate when computation scales beyond the training regime. To mitigate this and stabilize processing across many iterations, we introduce a discretization mechanism that *anchors* the model's latent representation as the number of recurrent iterations grows: after each iteration, the model's continuous hidden states are projected into a structured discrete symbolic space and immediately re-embedded to form the input for the next recurrent step. This ensures the input at each iteration lies in a shared structured space, even as computation scales beyond the training regime.

We implement this anchoring with a structured tokenization and embedding scheme, enabling each token's internal state to evolve recurrently while remaining grounded in a shared discrete space. In our task, the discrete latent space factorizes into four factors: token syntax, variable identity, numerical value, and operation type (see Appendix B.1 for a concrete example). We train separate embeddings for each factor and sum them to map discrete states back to distributed representations. At each iteration, we first apply the recurrent Transformer block, RecurrentTransformerBlock, as in Equation (2). We then discretize the resulting distributed representations via argmax decoding for each symbolic factor and re-embed the discrete

Table 1: *Guide to Implementation of Proposed Mechanisms in Baselines.* The leftmost column lists the method names, matching the figure legends. ● indicates that a mechanism is implemented, ○ indicates it is not, and ◐ signifies partial implementation.

| Method / Mechanism | Mechanism 1 | Mechanism 2 | Mechanism 3 | Mechanism 4 |
|---|---|---|---|---|
| *Feedforward End-to-End* | ○ | ○ | ○ | ○ |
| *Recurrent End-to-End* | ◐ | ○ | ○ | ○ |
| *Chain-of-Thought* | ◐ | ◐ | ○ | ○ |
| *Continuous Latent Space Supervision* | ● | ● | ○ | ○ |
| *Discrete Latent Space Supervision* | ● | ● | ● | ○ |
| *Discrete Latent Space Supervision ↻* | ● | ● | ● | ● |

state to form the input to the next iteration.

$$
\begin{aligned}
&\left[\tilde{E}_i^{(t+1)}\right]_{i=1}^n \leftarrow \text{RecurrentTransformerBlock}\left(\left[E_i^{(t)}\right]_{i=1}^n\right) \\
&z_{i,\{\texttt{factor}\}}^{(t+1)} \leftarrow \arg\max\{W_{\{\texttt{factor}\}}\,\tilde{E}_i^{(t+1)}\} \\
&E_{i,\{\texttt{factor}\}}^{(t+1)} \leftarrow \text{FactorEmbed}(z_{i,\{\texttt{factor}\}}^{(t+1)}) \\
&E_i^{(t+1)} \leftarrow \sum_{\texttt{factor}\in\mathcal{F}} E_{i,\{\texttt{factor}\}}^{(t+1)}.
\end{aligned}
\tag{4}
$$

Discrete latent states combine with input-adaptive recurrence (Mechanism 1) to yield a natural halting rule: iterate until a *fixed point* satisfying $\boldsymbol{z}^{(t+1)} = \boldsymbol{z}^{(t)}$. In a discrete state space, this implies $\boldsymbol{z}^{(t+k)} = \boldsymbol{z}^{(t)}$ for all $k \geq 0$, which helps avoid the "overthinking" behavior observed in prior work, where performance degrades when iterating far beyond the training regime [e.g., Ban+22]. Discreteness therefore provides a robust halting criterion based on *exact* equality of successive latent states, offering a simpler and more stable alternative to learned-halting approaches [e.g., Gra17; BBB21].

**Mechanism 4: Learning to Self-Correct.** Finally, we introduce a self-correction scheme to improve robustness over long computations, where errors are more likely to compound. During training, we inject small, structured perturbations into the discrete latent state (e.g., corrupting a subset of node values or intermediate states) and train the recurrent updates to repair them, encouraging attractor-like, self-correcting dynamics. Specifically, with small probability at each iteration, we randomly corrupt a subset of value components within the latent state, forcing the model to detect and correct incorrect values (whether induced by corruption or by the model itself) before proceeding.

## 4.2 Experimental Results & Discussion

Combining these mechanisms yields an architecture capable of effectively generalizing far beyond the training distribution to much larger and more complex inputs. As described in Section 3, we train on circuits with $N \leq 32$ and evaluate on progressively larger circuits up to $N = 128$. To isolate the effects of our proposed mechanisms, we study a collection of methods implementing different subsets of the mechanisms (see Table 1).

***Enabling Robust Algorithmic OOD Generalization.*** Figure 3a shows OOD generalization across ablations of the mechanisms, alongside the *Chain-of-Thought* and *End-to-End* baselines. The *End-to-End* models (both feedforward and recurrent) fail to learn the task beyond small circuit sizes—even in-distribution, under our stringent "fully solved" metric—although recurrence helps slightly. *Chain-of-Thought* supervision yields a substantial improvement, achieving near-perfect *in-distribution* performance ($N \leq 32$) and a limited degree of out-of-distribution generalization. To assess our mechanisms for robust OOD generalization, we evaluate three classes of models incorporating increasing subsets of these mechanisms. OOD performance improves sharply as more mechanisms are incorporated. When all mechanisms are included, i.e., *Discrete Latent Space Supervision ↻*[1], the model achieves *near-perfect performance* across all OOD splits we examined.

---

[1]Here, the ↻ symbol represents self-correction.

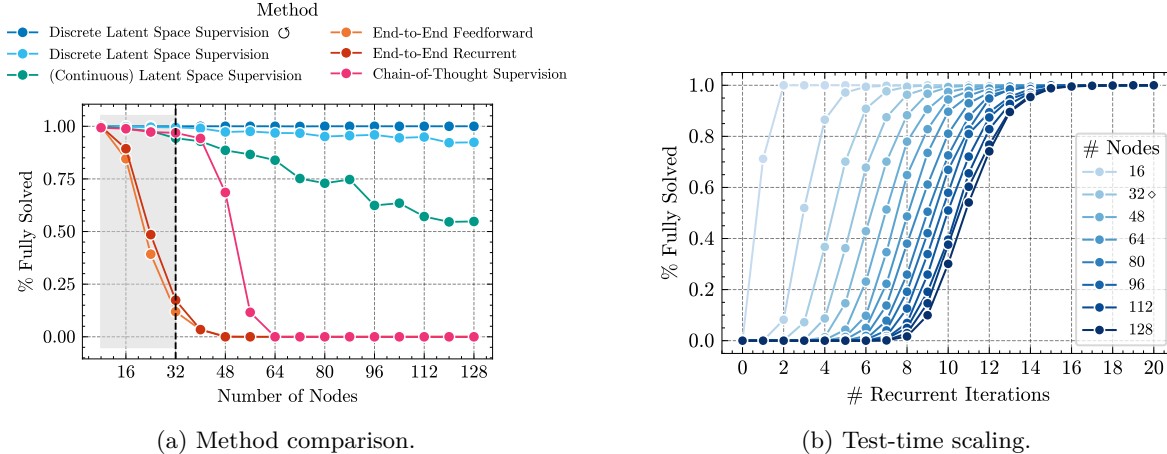

(a) Method comparison.

(b) Test-time scaling.

Figure 3: Empirical results for robust out-of-distribution generalization. **(a)** Out-of-distribution generalization performance across methods on the modular arithmetic circuits task. **(b)** Input-adaptive scaling of computation time for *Discrete Latent Space Supervision* ↺.

***Depth-Invariance for Scalable Reasoning.*** Generalizing beyond the training regime requires scaling computation with input complexity. Chain-of-thought (CoT) does so by lengthening an autoregressive trace, but forces computation into token-by-token text rather than the model's native latent space, which can be inefficient and brittle. Our recurrent formulation instead scales by iterating the same recurrent block, imposing a depth-invariant structure: the model learns a step-wise update rule that can be applied for many more iterations than seen in training. This parallels invariances in geometric deep learning [Bro+17; Bro+21; Ger+23]. Here, recurrence imposes invariance under the network's own iterative action, yielding a scalable, recursive algorithm capable of solving much larger, more complex instances.

***The Importance of Anchored Discrete Representations.*** In Figure 3a, *Continuous Latent Space Supervision* is a recurrent model with step-by-step algorithmic supervision of continuous latent states. Unlike *Discrete Latent Space Supervision*, it does not discretize the latent states between recurrent iterations. While this outperforms the *Chain-of-Thought* baseline (which is limited to linear reasoning paths), its out-of-distribution performance degrades on larger inputs that require increasing recurrent depth and computation time. We attribute this to noise accumulation in continuous representations—exacerbated by scaling test-time compute—which can cause drift from the semantically meaningful manifold learned during training. *Discrete Latent Space Supervision* retains step-by-step supervision (Mechanism 2) but discretizes between iterations (Mechanism 3). This "anchors" the latent states to a semantically consistent representation space, enabling deeper computation without noise accumulation and yielding markedly improved OOD generalization.

***Error-Correction Leads to Greater Robustness in Scaling.*** In *Discrete Latent Space Supervision* ↺, we add explicit supervision for error correction by randomly corrupting the model's latent space with some small probability during training. While the model may make occasional errors, it often corrects them in the next recurrent iteration, yielding near-perfect OOD generalization. Interestingly, we find that error correction requires more layers in the recurrent block in order to succeed. An intuitive explanation is that error correction requires greater computational depth *per step*: the model must first identify and correct errors from prior steps before executing the current step's computation.

***Robust Test-time Scaling.*** Many tasks require computation that grows with input size or complexity, so solving larger instances than those seen in training requires scaling compute beyond the training regime. In our setting, where the reasoning process is latent, we do this by increasing the number of recurrent iterations. Figure 3b shows the fraction of instances solved as a function of iteration count: more iterations enable solving progressively larger and harder instances, and our mechanisms enable robust scaling.

***Details, Extensions & Further Ablations.*** The appendices provide additional discussion, results, and hyperparameter-search details. Positional encoding and model depth matter substantially, so for each method we report the best model found; for CoT baselines, we also report the best of several reasoning-chain designs.

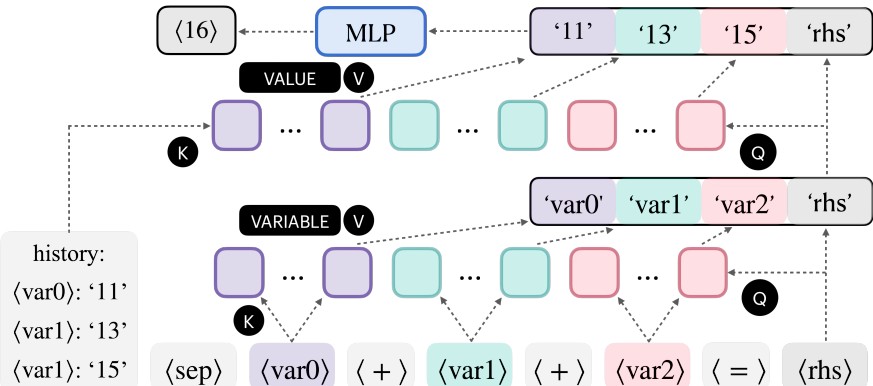

Figure 4: Illustration of the two-layer model performing the modular addition task. Colored squares represent attention heads, grouped by the variable positions they attend to. Black rectangles indicate the embedding components chosen by the value projection matrix. $\langle \cdot \rangle$ denotes tokens, and '·' denotes embedding components.

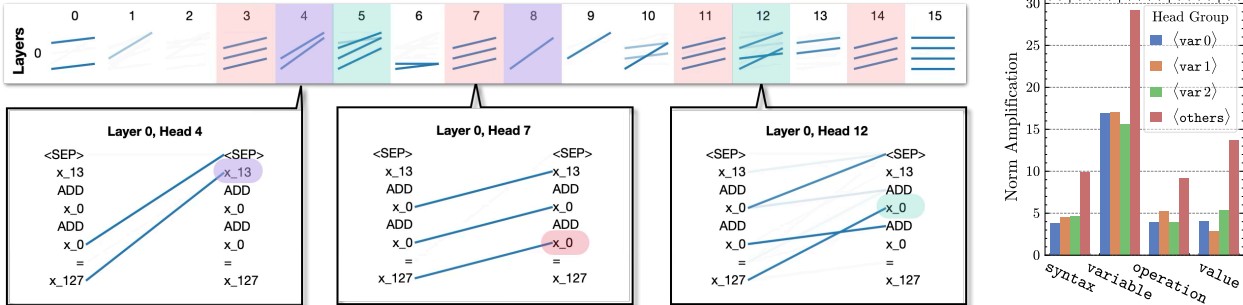

Figure 5: **Left.** An illustration of the functionality of attention heads in the first layer. Heads 4 and 8 attend to the first variable position, Heads 5 and 12 attend to the second variable position, Heads 3, 7, 11, and 14 attend to the third variable position, and the remaining heads attend to the RHS position or do not show a clear attention pattern. **Right.** Norm amplification of each factor's embeddings passed through the combined attention OV matrix by head groups. $\langle \text{others} \rangle$ exhibits significantly higher norm amplification, primarily because Head 15 performs a self-copy operation at the RHS position.

We next analyze the model mechanistically, mapping the computational circuits implemented by each component and connecting these circuits to the mechanisms we propose to explain robust OOD generalization.

## 5    Mechanistic Interpretability

In this section, we aim to answer the following questions via a detailed study of the model's inner workings: *(i)* What algorithm does the trained model implement? *(ii)* Why is the trained model able to generalize to OOD data? To answer these questions, we first propose hypotheses on the functionality of each model block: first-layer attention, second-layer attention, and the final MLP. For each of these hypotheses, we conduct controlled experiments where we apply causal interventions to specific parts of the input and isolate the effect on model activations to identify the function of each component. Our methodology builds on prior work on causal interpretability in neural networks [Gei+21; Men+22; Gei+24], but is tailored to interpreting *recurrent* transformer models. Complete experimental details are in the appendix.

**Induction Head & Modular Addition Mechanism**

To understand the algorithm implemented by the trained model, we analyze in detail the recurrent Transformer model trained with our proposed *Discrete Latent Space Supervision* method on the mathematical reasoning task. The recurrent Transformer model is configured with two layers, 16 attention heads, and a hidden state dimension of 256. For more details on the model configuration, please refer to Appendix D. We summarize

our mechanism analysis results in Figure 4, where we reveal an induction head mechanism operating within the two-layer attention block and a modular addition mechanism in the final feedforward layer. To ground the discussion, consider the example equation

$$[\texttt{sep}] \ \langle\texttt{var0}\rangle \ \langle+\rangle \ \langle\texttt{var1}\rangle \ \langle+\rangle \ \langle\texttt{var2}\rangle \ \langle=\rangle \ \langle\texttt{rhs}\rangle \, .$$

We decompose the model's computation at the RHS into three components. In layer 1, attention heads copy the "`variable`" factored embeddings of $\langle\texttt{var0}\rangle$, $\langle\texttt{var1}\rangle$, and $\langle\texttt{var2}\rangle$ to the RHS. In layer 2, attention heads use these names to retrieve their values from prior equations via an induction head. The final feedforward layer then adds these values and writes the sum to the RHS.

**First Layer Attention Performs Variable Copying.** The attention heads in the first layer are grouped by the variable position they attend to, reflecting an attention pattern that is dependent on relative position, as illustrated in Figure 5 (left). For the token embeddings of $\langle\texttt{var0}\rangle$, $\langle\texttt{var1}\rangle$, and $\langle\texttt{var2}\rangle$, which comprise four separate factored embedding types (`syntax`, `variable`, `operation`, and `value`), the value and output projection matrices of each head group *select a subspace* of these token embeddings containing only the `variable` embeddings. This is evident in Figure 5 (right), which plots the norm amplification for different factored embedding types. More details on the norm amplification calculation can be found in Appendix D. This shows that the first layer attention copies the *variable names* of its parents, which will later be used to obtain their values in the second layer.

**Second Layer Attention Implements Variable-Dependent Induction Head Mechanism.** The second layer's attention heads retrieve the values of $\langle\texttt{var0}\rangle$, $\langle\texttt{var1}\rangle$, and $\langle\texttt{var2}\rangle$ from previous equations via an induction-head mechanism [Ols+22]. The heads are grouped by which variable value they retrieve. For example, suppose one group is responsible for retrieving $\langle\texttt{var0}\rangle$. The heads in this group attend to the first occurrence of $\langle\texttt{var0}\rangle$—the RHS of a previous equation, where $\langle\texttt{var0}\rangle$ is first computed—and copy its "`value`" factored embedding to the current RHS position. In summary, the variable names copied in the first layer serve as queries to retrieve these variables' values by searching over RHS positions in previous equations.

**Feedforward Layer Performs Modular Addition.** The last feedforward layer implements a modular addition mechanism, where the model computes the sum of the values of the variables on the LHS and outputs the result to the RHS position. There are many works that have studied how the feedforward layer implements a modular addition mechanism in the context of Transformer networks [Nan+23; Tia24; Dos+24] using a Fourier-based approach. We provide additional evidence for this mechanism in Appendix D.

***OOD Generalization of the Trained Model.*** The model's robust OOD generalization stems from its architectural mechanisms, which guide it towards learning a universal and scalable algorithm. In particular, it learns a variable-dependent induction head mechanism that is invariant to length, leveraging both relative-positional and variable-dependent attention patterns, which enables the model to operate over contexts of arbitrary lengths. Thus, despite being trained on circuits with limited size, the input-adaptive recurrence, intermediate supervision, and discretization mechanisms enable the model to learn a scalable algorithm capable of solving problems of increased complexity.

## 6 Discussion

### 6.1 Architectural ingredients for depth-invariant algorithmic generalization

This work identifies a compact set of architectural ingredients that enable strong out-of-distribution algorithmic generalization: latent recurrent computation (Mechanism 1), algorithmic alignment supervision (Mechanism 2), discrete latent representations (Mechanism 3), and error-correcting dynamics (Mechanism 4). Together, these mechanisms support scalable and robust iterative computation in latent space, enabling the model to scale its effective computational depth in proportion to input complexity. Our results show that these ingredients form a minimal recipe for learning depth-invariant algorithms in neural networks. Unlike token-level Chain-of-Thought, which relies on explicit reasoning traces, latent recurrence reasons internally and avoids autoregressive token-generation constraints. The approach complements CoT-style supervision by providing architectural inductive bias rather than depending solely on data-driven pattern learning.

### 6.2 Limitations and future work

The main limitation of this study is that we validate our proposal on a single task domain. This design choice enabled a level of experimental control and mechanistic analysis difficult to achieve on broader benchmarks: we could precisely manipulate instance complexity, fully isolate the contribution of each mechanism, and perform mechanistic analysis. Such deeply controlled studies are often essential in the early stages of developing new algorithmic reasoning methods. Nevertheless, it remains to evaluate the full architecture, and its ablated variants, on additional tasks. Extending these mechanisms to richer settings such as program execution, dynamic programming lattices, or real-world mathematical reasoning benchmarks is a promising direction.

### 6.3 Beyond modular arithmetic circuits

We selected modular arithmetic circuits because they instantiate core structural properties of algorithmic computation while allowing fine-grained control over compositional depth. Any program over a finite set of primitive operations can be represented as a directed acyclic computation graph, or circuit; depth then corresponds to the number of sequential computational steps required for evaluation. Our circuits include branching, reconvergence, and reuse of intermediate values, reflecting the compositional challenges inherent in general algorithmic tasks (e.g., expression evaluation, symbolic manipulation, and dynamic programming). The task is therefore not specific to modular arithmetic: it represents a canonical instance of a much broader class of DAG-structured reasoning problems. Future work may explore variants involving different primitive operation sets, richer control-flow patterns, or partially observed graph structures, allowing systematic investigation of how each mechanism contributes across a spectrum of algorithmic tasks.

### 6.4 Generality of the recurrent latent-space architecture with discrete states

The four mechanisms introduced in this work operate at the level of generic DAG-structured computations and do not depend on the specifics of modular arithmetic. Input-adaptive recurrence (Mechanism 1) applies whenever the target algorithm requires variable computational depth, which is a common characteristic of algorithmic computation. Algorithm-alignment supervision (Mechanism 2) assumes only that the underlying computation admits an iterative decomposition, which is true for any DAG-structured algorithm; moreover, as shown in Appendix C, this structure can be learned implicitly without oracle-provided depths. Discrete latent states (Mechanism 3) fit naturally in domains where intermediate values lie in a finite symbolic space and provide the stability needed for long-chain computation and fixed-point halting. Finally, error correction (Mechanism 4) is broadly applicable: perturbing intermediate states to encourage self-correcting dynamics is a task-agnostic strategy for preventing error accumulation. Together, these mechanisms define a general architectural template for depth-invariant algorithmic reasoning.

### 6.5 Integration with language models

A natural question is how latent recurrent reasoning with discrete states might integrate with large-scale language models. Autoregressive Transformers reason through token generation, whereas our architecture reasons in latent space through iterative refinement. Combining these paradigms may yield hybrid models in which latent recurrence supports algorithmic stability and input-adaptive computation, while continuous representations and autoregressive decoding preserve the flexibility and generality of modern language models. Potential avenues include embedding a discrete latent recurrent module in a pretrained model, introducing hybrid continuous–discrete latent spaces, or using latent computation as an internal reasoning step between forward passes. As LLMs are increasingly applied to tasks requiring deeper compositional generalization, the mechanisms studied in this work may provide useful inductive biases for stabilizing long-chain reasoning.

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

             2402.09371 [cs]. Pre-published (cited on pages 1, 2, 6).

# A    Experimental Details on Chain-of-Thought & End-to-End Baselines

This section provides further experimental details on the chain-of-thought and end-to-end baselines.

## A.1    End-to-End Baselines

The end-to-end models in our experiments are causal encoder-only Transformer models with a fixed depth and/or number of iterations that are trained with end-to-end supervision only. That is, they receive supervision on the final solution, but do not receive fine-grained supervision on the intermediate steps to explicitly align the models to a universal algorithmic problem-solving procedure.

Within the end-to-end baselines, we consider feedforward models and recurrent models. Feedforward models have a fixed number of layers and independently-learned parameters at each layer. Recurrent models, on the other hand, have a recurrent block consisting of some number of Transformer layers, which is applied recurrently for a fixed number of iterations.

Recognizing the importance of positional encoding for length generalization [Kaz+23], we explore several positional encoding methods for each class of methods that we evaluate. In particular, we evaluate learned absolute positional embeddings [Vas+17] (AbPE), Rotary Positional Encoding [Su+23] (RoPE), No Positional Encoding [Kaz+23] (NoPE), and the relative positional-encoding method proposed by [He+21] (DeBERTa).

We perform a hyperparameter search across each of these factors, varying the number of recurrent iterations $T$, the number of layers per recurrent block $L$, the hidden state dimension $D$, and the positional encoding method. As described in the main text, we train on a dataset of examples with up to 32 nodes, and evaluate on examples varying in size from 8 nodes to 128 nodes. Figure 6 depicts the average OOD performance as measured by the "% Fully Solved" metric for each baseline model configuration. The results in the main text correspond to the best-performing end-to-end models according to this metric. In particular, the best-performing recurrent model is RoPE-T4L2H16D256, and the best-performing feedforward model is DeBERTa-T1L8H16D256. Note that the naming scheme describes the positional encoding method, the number of recurrent steps $T$, the number of layers $L$ in the Transformer block, the number of attention heads $H$, and the model dimension $D$. $T = 1$ corresponds to a "feedforward" model with no recurrence.

Figure 7 depicts additional experimental results for the end-to-end baseline experiments.

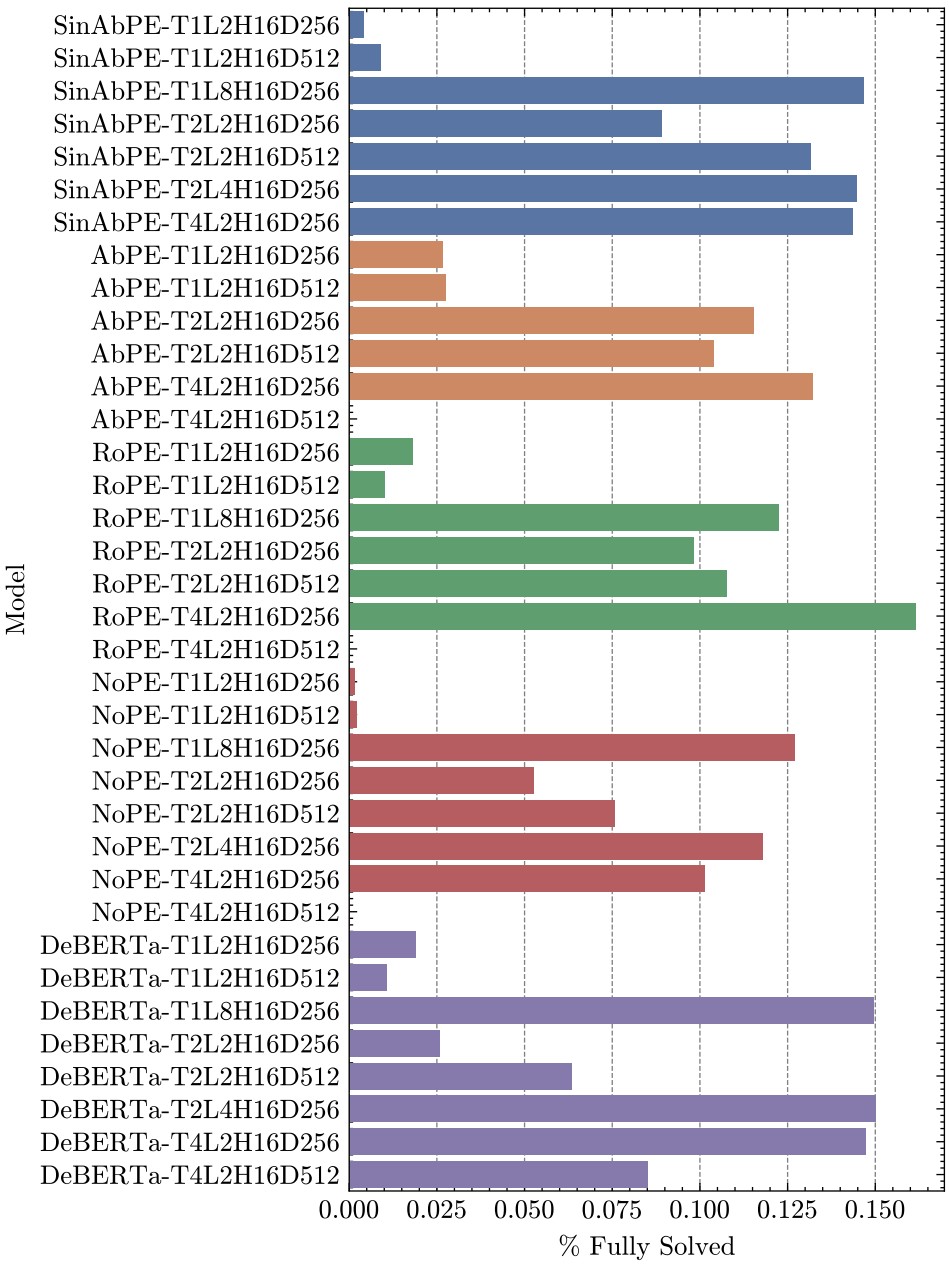

Figure 6: A comparison of average OOD generalization performance of different feedforward and recurrent baselines, varying architectural hyperparameters. This is computed as the average of the "% Fully Solved" metric computed on inputs of varying size from $N = 8$ to $N = 128$.

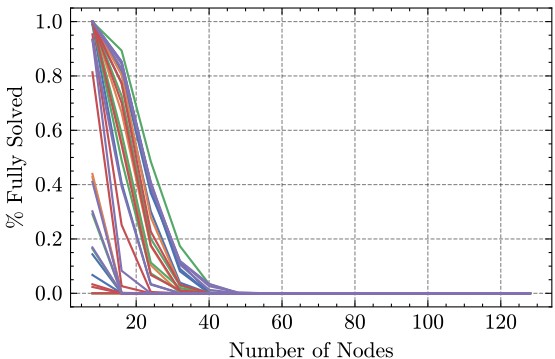

(a) Each line corresponds to an experimental run. Lines are color-coded by positional encoding, but other architectural hyperparameters vary and are not represented.

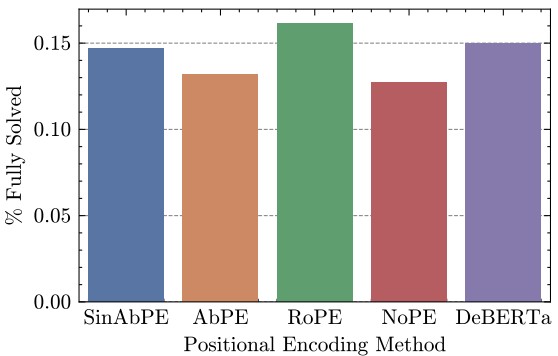

(b) Average "% Fully Solved" across test splits for the best model of each positional encoding method. The relative positional encoding methods (RoPE and DeBERTa) perform best.

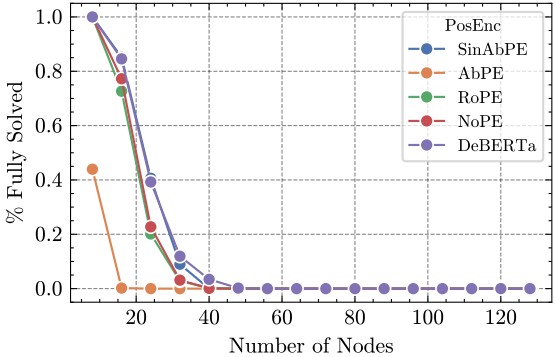

(c) % Fully solved by graph size for best model of each positional encoding method in the *feedforward* baselines.

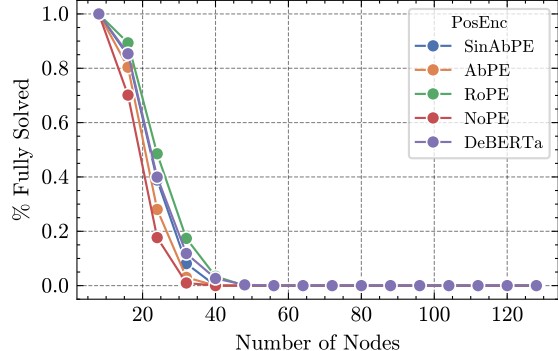

(d) % Fully solved by graph size for best model of each positional encoding method in the *recurrent* baselines.

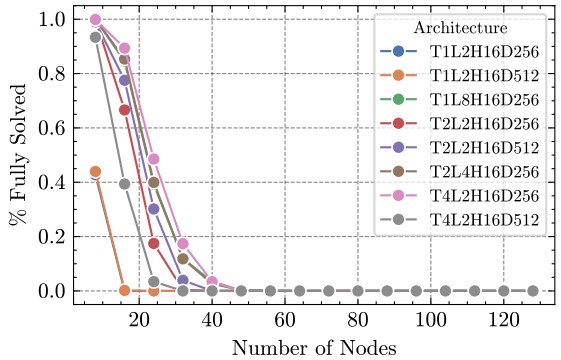

(e) % Fully solved by graph size for the best model of each architectural configuration. Recurrent models slightly outperform feedforward models. Computational depth (i.e., $T \cdot L$) is crucial, with shallow models performing poorly even on the smallest in-distribution inputs.

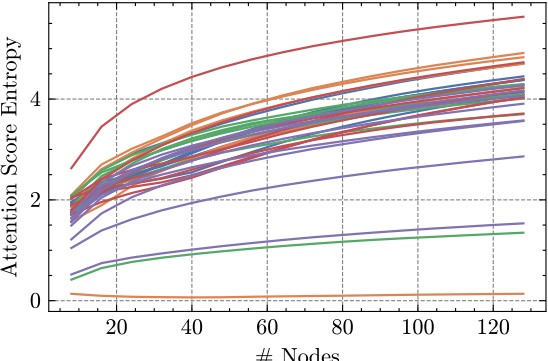

(f) Average attention score entropy by input size. Attention scores disperse as the input size increases.

Figure 7: Further experimental results for end-to-end baselines. All end-to-end models struggle to generalize beyond the training distribution, regardless of architectural hyperparameters.

## A.2 Chain-of-Thought Baselines

The chain-of-thought baselines in our experiments are causal Transformer language models that are trained with a next-token prediction objective on sequence data that includes a step-by-step solution of the problem instance. The models are evaluated by prompting them with the problem instance and autoregressively generating the entire chain-of-thought via a greedy decoding procedure.

We begin by providing more details on the construction of the chain-of-thought trajectories for these baselines, then provide further details on the experimental setup and present additional results.

### A.2.1 Chain-of-Thought Trajectories

We experiment with a few different types of chain-of-thought trajectories, providing different levels and styles of supervision on the intermediate computation.

As described in the main text, the first part of the sequence is always the description of the input problem, which matches the format of the other methods we consider: a sequence of equations that define a computational graph to be solved. This is then followed by a special $\langle \texttt{CoT} \rangle$ token which indicates the end of the input and the beginning of the chain-of-thought. The chain-of-thought involves solving each variable in the input in linear order, one-by-one.

We experiment with two types of CoT trajectories that vary the level of detail. The first provides supervision on the values only. The CoT simply recalls that variables one-by-one and computes their values, without recalling the equation that defined them.

$$[...\text{Input Prompt}...]\langle \texttt{CoT} \rangle[...]\langle x_{101} \rangle \langle = \rangle \langle 4 \rangle$$

The second type of CoT trajectory involves first recalling the equation that defined the variable, then recalling the values of the variables in the equation, and then computing the value of the desired variable. This requires a longer chain-of-thought but provides richer supervision.

$$[...\text{Input Prompt}...]\langle \texttt{CoT} \rangle[...]\langle x_{101} \rangle \langle = \rangle \langle x_{23} \rangle \langle + \rangle \langle x_{91} \rangle \langle = \rangle \langle 22 \rangle \langle + \rangle \langle 5 \rangle \langle = \rangle \langle 4 \rangle$$

Below, we provide an example of a *full* CoT trajectory on an input with $N = 32$ nodes.

*Add corresponding graph for this*

$\langle 2\rangle\langle =\rangle\langle x_3\rangle\,[\mathsf{sep}]\,\langle 2\rangle\langle =\rangle\langle x_{30}\rangle\,[\mathsf{sep}]\,\langle 18\rangle\langle =\rangle\langle x_{12}\rangle\,[\mathsf{sep}]\,\langle 14\rangle\langle =\rangle\langle x_{11}\rangle\,[\mathsf{sep}]$
$\langle 15\rangle\langle =\rangle\langle x_{20}\rangle\,[\mathsf{sep}]\,\langle 8\rangle\langle =\rangle\langle x_{23}\rangle\,[\mathsf{sep}]\,\langle x_{30}\rangle\langle =\rangle\langle x_9\rangle\,[\mathsf{sep}]\,\langle x_{23}\rangle\langle +\rangle\langle x_3\rangle\langle =\rangle\langle x_{22}\rangle\,[\mathsf{sep}]$
$\langle x_{20}\rangle\langle \times\rangle\langle x_{23}\rangle\langle =\rangle\langle x_{27}\rangle\,[\mathsf{sep}]\,\langle x_3\rangle\langle +\rangle\langle x_{22}\rangle\langle =\rangle\langle x_0\rangle\,[\mathsf{sep}]\,\langle x_3\rangle\langle +\rangle\langle x_{22}\rangle\langle \times\rangle\langle x_{11}\rangle\langle =\rangle\langle x_{26}\rangle\,[\mathsf{sep}]$
$\langle x_{20}\rangle\langle -\rangle\langle x_{22}\rangle\langle +\rangle\langle x_{23}\rangle\langle =\rangle\langle x_{13}\rangle\,[\mathsf{sep}]\,\langle x_{22}\rangle\langle =\rangle\langle x_{24}\rangle\,[\mathsf{sep}]\,\langle x_{12}\rangle\langle \times\rangle\langle x_{23}\rangle\langle -\rangle\langle x_0\rangle\langle =\rangle\langle x_{17}\rangle\,[\mathsf{sep}]$
$\langle x_{11}\rangle\langle \times\rangle\langle x_{26}\rangle\langle =\rangle\langle x_{28}\rangle\,[\mathsf{sep}]\,\langle x_{13}\rangle\langle -\rangle\langle x_{11}\rangle\langle +\rangle\langle x_{23}\rangle\langle =\rangle\langle x_{21}\rangle\,[\mathsf{sep}]\,\langle x_{17}\rangle\langle -\rangle\langle x_3\rangle\langle =\rangle\langle x_{25}\rangle\,[\mathsf{sep}]$
$\langle x_{30}\rangle\langle \times\rangle\langle x_{17}\rangle\langle -\rangle\langle x_{23}\rangle\langle =\rangle\langle x_6\rangle\,[\mathsf{sep}]\,\langle x_{17}\rangle\langle =\rangle\langle x_{16}\rangle\,[\mathsf{sep}]\,\langle x_{11}\rangle\langle +\rangle\langle x_{21}\rangle\langle =\rangle\langle x_7\rangle\,[\mathsf{sep}]$
$\langle x_{28}\rangle\langle +\rangle\langle x_{17}\rangle\langle -\rangle\langle x_{21}\rangle\langle =\rangle\langle x_{14}\rangle\,[\mathsf{sep}]\,\langle x_7\rangle\langle =\rangle\langle x_{15}\rangle\,[\mathsf{sep}]\,\langle x_7\rangle\langle =\rangle\langle x_{31}\rangle\,[\mathsf{sep}]$
$\langle x_{12}\rangle\langle +\rangle\langle x_3\rangle\langle +\rangle\langle x_{14}\rangle\langle =\rangle\langle x_5\rangle\,[\mathsf{sep}]\,\langle x_{14}\rangle\langle =\rangle\langle x_{19}\rangle\,[\mathsf{sep}]\,\langle x_{23}\rangle\langle -\rangle\langle x_5\rangle\langle \times\rangle\langle x_7\rangle\langle =\rangle\langle x_{29}\rangle\,[\mathsf{sep}]$
$\langle x_5\rangle\langle =\rangle\langle x_{18}\rangle\,[\mathsf{sep}]\,\langle x_{25}\rangle\langle +\rangle\langle x_{23}\rangle\langle -\rangle\langle x_{19}\rangle\langle =\rangle\langle x_4\rangle\,[\mathsf{sep}]\,\langle x_{14}\rangle\langle \times\rangle\langle x_{29}\rangle\langle -\rangle\langle x_5\rangle\langle =\rangle\langle x_2\rangle\,[\mathsf{sep}]$
$\langle x_{29}\rangle\langle \times\rangle\langle x_{28}\rangle\langle -\rangle\langle x_7\rangle\langle =\rangle\langle x_1\rangle\,[\mathsf{sep}]\,\langle x_3\rangle\langle \times\rangle\langle x_{23}\rangle\langle \times\rangle\langle x_{18}\rangle\langle =\rangle\langle x_8\rangle\,[\mathsf{sep}]$
$\langle x_8\rangle\langle -\rangle\langle x_{28}\rangle\langle -\rangle\langle x_0\rangle\langle =\rangle\langle x_{10}\rangle\,\langle \mathtt{CoT}\rangle\,\langle x_3\rangle\langle =\rangle\langle 2\rangle\,[\mathsf{sep}]\,\langle x_{30}\rangle\langle =\rangle\langle 2\rangle\,[\mathsf{sep}]$
$\langle x_{12}\rangle\langle =\rangle\langle 18\rangle\,[\mathsf{sep}]\,\langle x_{11}\rangle\langle =\rangle\langle 14\rangle\,[\mathsf{sep}]\,\langle x_{20}\rangle\langle =\rangle\langle 15\rangle\,[\mathsf{sep}]\,\langle x_{23}\rangle\langle =\rangle\langle 8\rangle\,[\mathsf{sep}]$
$\langle x_9\rangle\langle =\rangle\langle x_{30}\rangle\langle =\rangle\langle 2\rangle\,[\mathsf{sep}]\,\langle x_{22}\rangle\langle =\rangle\langle x_{23}\rangle\langle +\rangle\langle x_3\rangle\langle =\rangle\langle 10\rangle\,[\mathsf{sep}]\,\langle x_{27}\rangle\langle =\rangle\langle x_{20}\rangle\langle \times\rangle\langle x_{23}\rangle\langle =\rangle\langle 5\rangle\,[\mathsf{sep}]$
$\langle x_0\rangle\langle =\rangle\langle x_3\rangle\langle +\rangle\langle x_{22}\rangle\langle =\rangle\langle 12\rangle\,[\mathsf{sep}]\,\langle x_{26}\rangle\langle =\rangle\langle x_3\rangle\langle +\rangle\langle x_{22}\rangle\langle \times\rangle\langle x_{11}\rangle\langle =\rangle\langle 7\rangle\,[\mathsf{sep}]$
$\langle x_{13}\rangle\langle =\rangle\langle x_{20}\rangle\langle -\rangle\langle x_{22}\rangle\langle +\rangle\langle x_{23}\rangle\langle =\rangle\langle 13\rangle\,[\mathsf{sep}]\,\langle x_{24}\rangle\langle =\rangle\langle x_{22}\rangle\langle =\rangle\langle 10\rangle\,[\mathsf{sep}]$
$\langle x_{17}\rangle\langle =\rangle\langle x_{12}\rangle\langle \times\rangle\langle x_{23}\rangle\langle -\rangle\langle x_0\rangle\langle =\rangle\langle 17\rangle\,[\mathsf{sep}]\,\langle x_{28}\rangle\langle =\rangle\langle x_{11}\rangle\langle \times\rangle\langle x_{26}\rangle\langle =\rangle\langle 6\rangle\,[\mathsf{sep}]$
$\langle x_{21}\rangle\langle =\rangle\langle x_{13}\rangle\langle -\rangle\langle x_{11}\rangle\langle +\rangle\langle x_{23}\rangle\langle =\rangle\langle 7\rangle\,[\mathsf{sep}]\,\langle x_{25}\rangle\langle =\rangle\langle x_{17}\rangle\langle -\rangle\langle x_3\rangle\langle =\rangle\langle 15\rangle\,[\mathsf{sep}]$
$\langle x_6\rangle\langle =\rangle\langle x_{30}\rangle\langle \times\rangle\langle x_{17}\rangle\langle -\rangle\langle x_{23}\rangle\langle =\rangle\langle 3\rangle\,[\mathsf{sep}]\,\langle x_{16}\rangle\langle =\rangle\langle x_{17}\rangle\langle =\rangle\langle 17\rangle\,[\mathsf{sep}]$
$\langle x_7\rangle\langle =\rangle\langle x_{11}\rangle\langle +\rangle\langle x_{21}\rangle\langle =\rangle\langle 21\rangle\,[\mathsf{sep}]\,\langle x_{14}\rangle\langle =\rangle\langle x_{28}\rangle\langle +\rangle\langle x_{17}\rangle\langle -\rangle\langle x_{21}\rangle\langle =\rangle\langle 16\rangle\,[\mathsf{sep}]$
$\langle x_{15}\rangle\langle =\rangle\langle x_7\rangle\langle =\rangle\langle 21\rangle\,[\mathsf{sep}]\,\langle x_{31}\rangle\langle =\rangle\langle x_7\rangle\langle =\rangle\langle 21\rangle\,[\mathsf{sep}]\,\langle x_5\rangle\langle =\rangle\langle x_{12}\rangle\langle +\rangle\langle x_3\rangle\langle +\rangle\langle x_{14}\rangle\langle =\rangle\langle 13\rangle\,[\mathsf{sep}]$
$\langle x_{19}\rangle\langle =\rangle\langle x_{14}\rangle\langle =\rangle\langle 16\rangle\,[\mathsf{sep}]\,\langle x_{29}\rangle\langle =\rangle\langle x_{23}\rangle\langle -\rangle\langle x_5\rangle\langle \times\rangle\langle x_7\rangle\langle =\rangle\langle 10\rangle\,[\mathsf{sep}]$
$\langle x_{18}\rangle\langle =\rangle\langle x_5\rangle\langle =\rangle\langle 13\rangle\,[\mathsf{sep}]\,\langle x_4\rangle\langle =\rangle\langle x_{25}\rangle\langle +\rangle\langle x_{23}\rangle\langle -\rangle\langle x_{19}\rangle\langle =\rangle\langle 7\rangle\,[\mathsf{sep}]$
$\langle x_2\rangle\langle =\rangle\langle x_{14}\rangle\langle \times\rangle\langle x_{29}\rangle\langle -\rangle\langle x_5\rangle\langle =\rangle\langle 9\rangle\,[\mathsf{sep}]\,\langle x_1\rangle\langle =\rangle\langle x_{29}\rangle\langle \times\rangle\langle x_{28}\rangle\langle -\rangle\langle x_7\rangle\langle =\rangle\langle 16\rangle\,[\mathsf{sep}]$
$\langle x_8\rangle\langle =\rangle\langle x_3\rangle\langle \times\rangle\langle x_{23}\rangle\langle \times\rangle\langle x_{18}\rangle\langle =\rangle\langle 1\rangle\,[\mathsf{sep}]\,\langle x_{10}\rangle\langle =\rangle\langle x_8\rangle\langle -\rangle\langle x_{28}\rangle\langle -\rangle\langle x_0\rangle\langle =\rangle\langle 6\rangle$

### A.2.2 Experimental Details & Additional Results

We perform a hyperparameter search varying: the number of recurrent iterations $T$, the number of layers per recurrent block $L$, the hidden state dimension $D$, and the positional encoding method. As described in the main text, we train on a dataset of examples with up to 32 nodes, and evaluate on examples varying in size from 8 nodes to 128 nodes. Figure 8 depicts the average OOD performance as measured by the "% Fully Solved" metric for each baseline model configuration. The results in the main text correspond to the best-performing CoT-supervised model according to this metric, which is the RoPE-T4L2H16D256 model.

Figure 10 depicts additional experimental results for the end-to-end baseline experiments. We highlight a few observations here:

- Figure 9 shows that some models are able to recall the equation structure correctly in their CoT, but are unable to robustly compute the values correctly. This suggests that a common source of error in the CoT baselines is the arithmetic computation, rather than copying equations from the input.

- As with the end-to-end baselines, the positional encoding method was critical for performance and length generalization. Among the methods we evaluated, we found NoPE to perform best, generalizing well to 40 nodes when trained on $N \leq 32$ nodes. The other positional encoding methods fail to generalize beyond the training regime. No method generalized robustly beyond 40 nodes.

- As with the end-to-end baselines, the computational depth of the model had a significant effect on performance. In particular, 4-layer models failed to learn the task well, but 8-layer models achieved good in-distribution performance and a limited degree of out-of-distribution generalization.

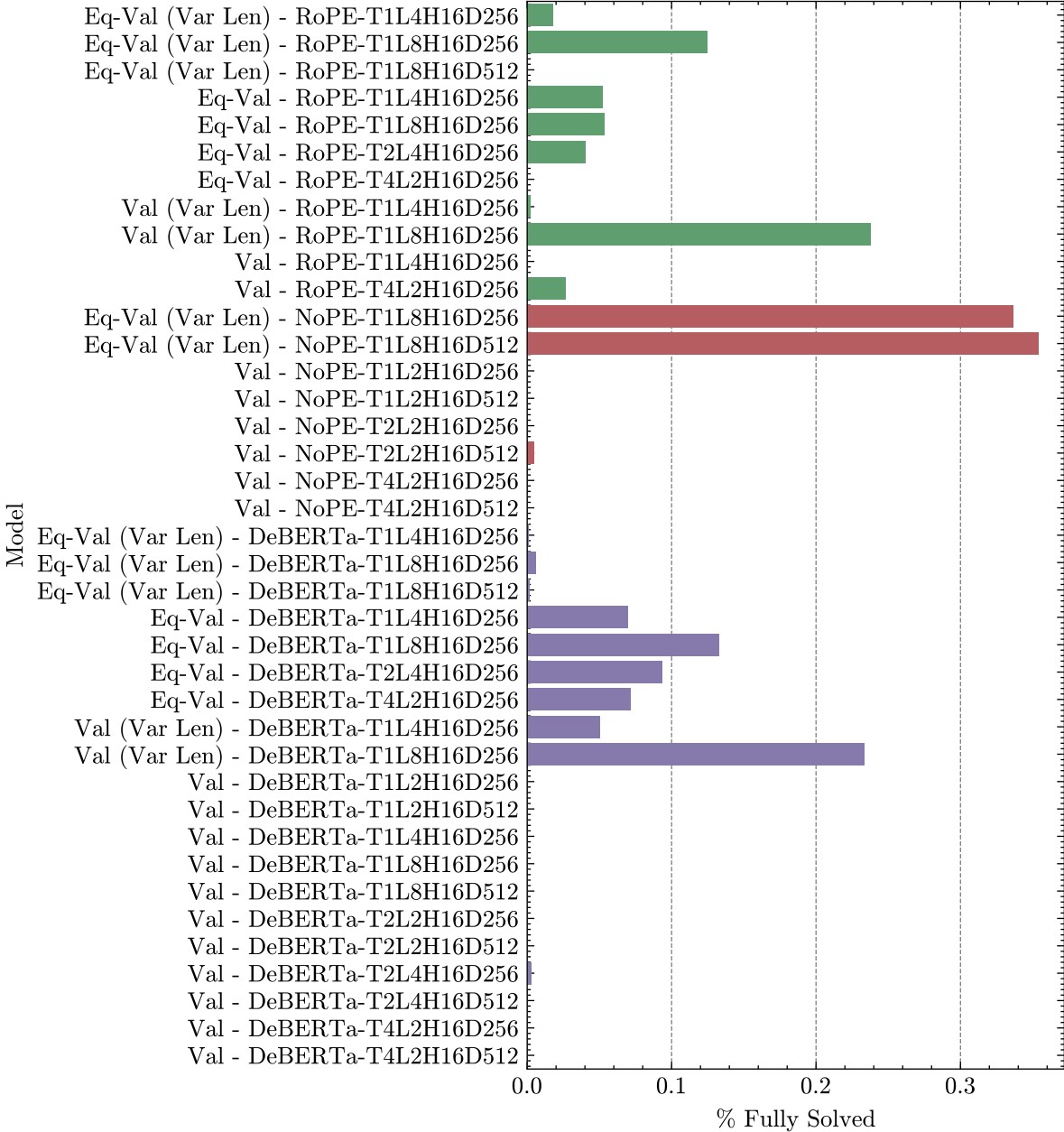

Figure 8: A comparison of average OOD generalization performance of different CoT-supervised baselines, varying architectural hyperparameters. The metric is full sequence accuracy, which measures the proportion of inputs where every node's value is computed correctly. The naming scheme matches the previous section, but adds a prefix describing the format of the CoT trajectories. "Val" means that the CoT trajectory directly computes the values of each variable, whereas "Eq-Val" first recalls the equations and then computes the values. Here, "(Var Len)" indicates runs where the input problem size is variable and randomly sampled in $N \leq 32$, rather than being only $N = 32$.

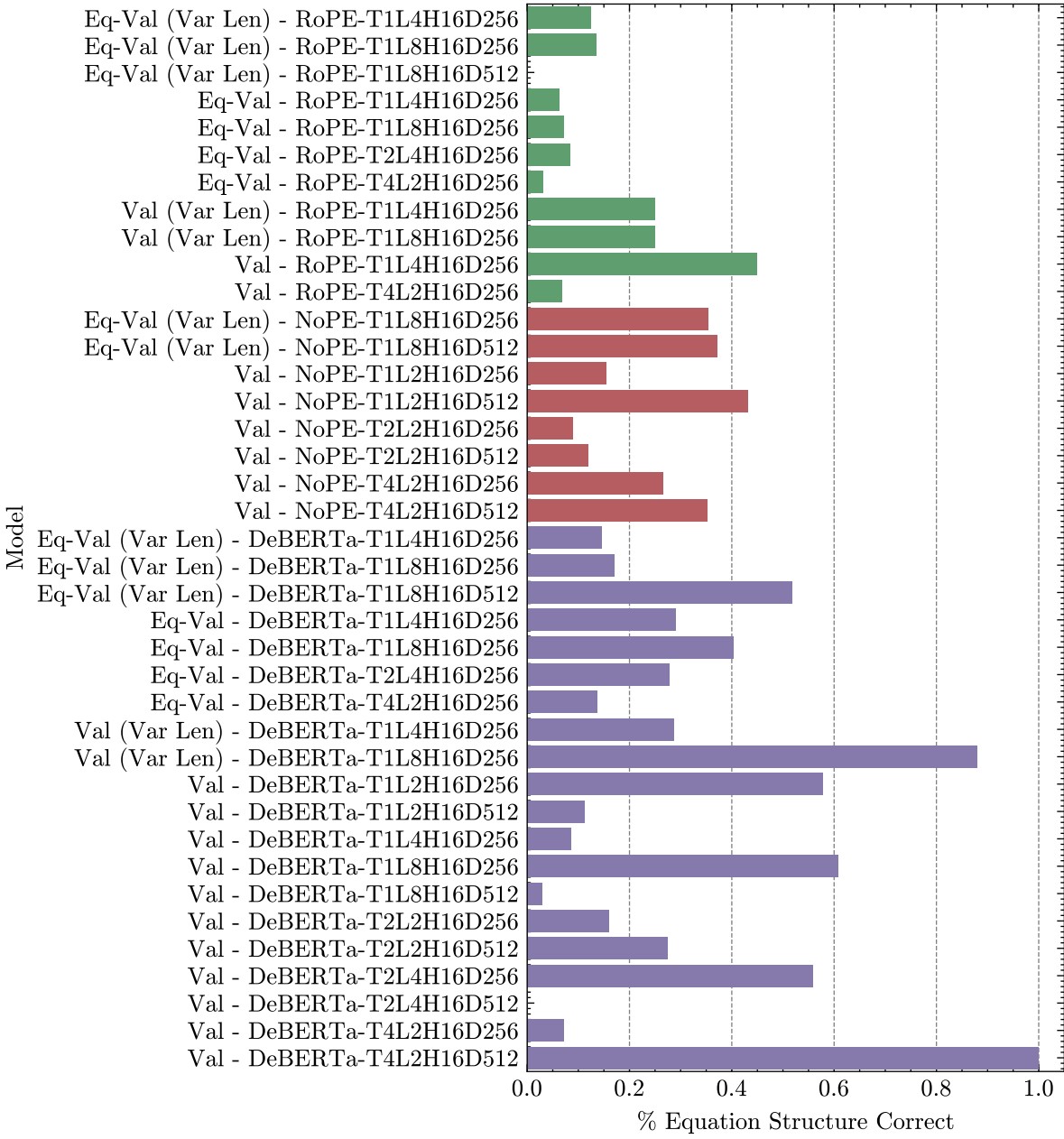

Figure 9: A comparison of average OOD generalization performance of different CoT-supervised baselines, varying architectural hyperparameters. The metric is "% Equation Structure Correct", which measures the proportion of inputs where the autoregressively generated CoT has the correct equation structure (without checking whether the values computed are correct).

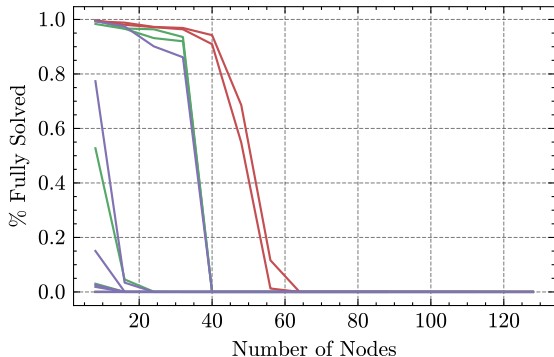

(a) Each line corresponds to an experimental run. Lines are color-coded by positional encoding, but other architectural hyperparameters vary and are not represented.

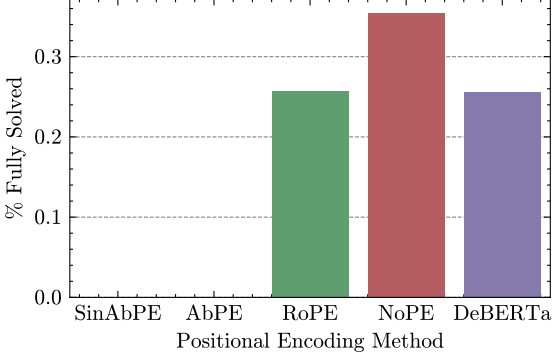

(b) Average OOD performance across test splits for the best model of each positional encoding method.

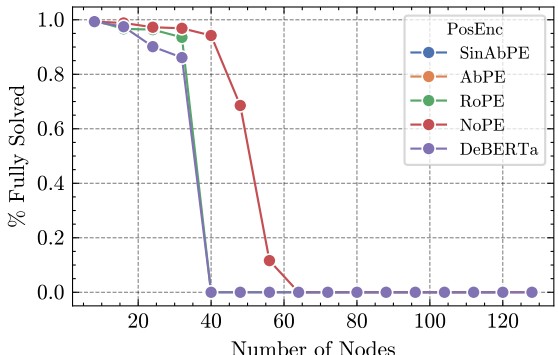

(c) % Fully solved by graph size for the best model of each positional encoding method. We find NoPE to achieve the best out-of-distribution generalization performance, generalizing well to 40 nodes when trained on $N \leq 32$ nodes. The other positional encoding methods fail to generalize beyond the training regime.

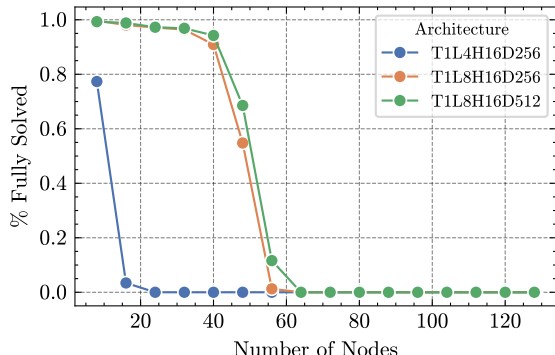

(d) % Fully solved by graph size for the best model of each architectural configuration. Computational depth (i.e., $T \cdot L$) is crucial for good performance, with shallow models performing poorly even in-distribution on larger inputs.

Figure 10: Further experimental results for CoT baselines. While chain-of-thought supervision yields improved performance over end-to-end models, out-of-distribution generalization capabilities are limited.

# B    Details on Latent State Supervision

## B.1    Latent State Embedding Structure

The input to the model is presented as a sequence of equations defining the value of each node in the computation graph. The vocabulary of the input includes variable names (e.g., $x_{42}$), numerical values (e.g., 17), operations (e.g., $+$), and special symbols like equality $\langle = \rangle$ or equation separation $[\texttt{sep}]$.

To provide the model with supervision on each part of the input, we employ a special tokenization and embedding scheme. We use a factored structure to tokenize each symbol in the input into 4-component tokens: syntax, variable, operation, and value. For example, the input $\langle 17 \rangle \langle = \rangle \langle x_{42} \rangle [\texttt{sep}] \ldots$ is tokenized as follows before the first iteration:

|  |  | syntax | variable | operation | value |
|---|---|---|---|---|---|
| $\langle 17 \rangle$ | $\rightarrow$ [ | value | N/A | N/A | 17 ] |
| $\langle = \rangle$ | $\rightarrow$ [ | $\langle = \rangle$ | N/A | N/A | N/A ] |
| $\langle x_{42} \rangle$ | $\rightarrow$ [ | variable | $x_{42}$ | N/A | empty ] |
| $[\texttt{sep}]$ | $\rightarrow$ [ | $[\texttt{sep}]$ | N/A | N/A | N/A ] |

The *syntax* factor can be `value`, `variable`, `operation`, or the special symbols $\langle = \rangle$ or $[\texttt{sep}]$. The *variable* factor is the variable names $\{x_0, \ldots, x_{127}\}$. The *operation* factor is the set of arithmetic operations (e.g., $+, -, \times$). The *value* factor is the set of numerical values (i.e., $\{0, \ldots, 22\}$). We also include an N/A symbol for the *variable*, *operation*, and *value* factors. For example, symbols with value syntax do not have a variable factor, etc. We also include a special `empty` symbol for the value factor of variable tokens. In the input to the model, the variable tokens have empty value factors because their values have not been computed yet. As the model processes the input, it iteratively computes the values of different variables and fills in their value factor.

We train a separate embedder for each factor, and map the input to vector embeddings by embedding each factor and adding the embeddings.

## B.2    Latent State Supervision

The *Continuous Latent Space Supervision*, *Discrete Latent Space Supervision*, and *Discrete Latent Space Supervision* ↻ methods share the same latent state supervision scheme. We train these recurrent models to learn to solve the input problem by computing node values one layer deeper in the computation graph with each recurrent iteration. We do this by defining a loss function at each iteration that penalizes predictions only for variables with depth less than or equal to the current iteration.

For each $\texttt{factor} \in \{\texttt{syntax}, \texttt{variable}, \texttt{operation}, \texttt{value}\}$, we learn a linear read-out layer $W_{\texttt{factor}} \in \mathbb{R}^{d_{\text{model}} \times |\mathcal{V}_{\texttt{factor}}|}$ that maps the vector state at the end of the recurrent iteration to a prediction of each factor. Here, $\mathcal{V}_{\texttt{factor}}$ denotes the vocabulary for the given factor (e.g., for the *value* factor, this is $\{0, \ldots, 22, \texttt{N/A}, \texttt{empty}\}$).

We provide the model with supervision on its latent states by defining a loss for each factor and at each recurrent iteration. In particular, the loss function for the value factor is defined such that the model is trained to predict the values of all variables that occur at depth $\leq t$ in the computation graph. In particular, for an input sequence $X = (x_1, \ldots, x_n)$, the value factor loss at iteration $t$ is defined as

$$\text{Loss}(\texttt{factor} = \texttt{value}, \text{iteration} = t) = \sum_{\substack{i \in [n] \\ \text{Depth}(x_i) \leq t}} \ell\left(W_{\texttt{value}} E_i^{(t)}, \text{Value}(x_i)\right). \tag{5}$$

where $\text{Depth}(x_i)$ is the depth of the variable $x_i$ in the input computation graph, $\text{Value}(x_i)$ is its computed value, and $E_i^{(t)} \in \mathbb{R}^{d_{\text{model}}}$ is the vector embedding of $x_i$ at recurrent iteration $t$. Here, $\ell$ is the cross-entropy loss.

The overall loss used to train the models is the sum of the individual factor losses at each iteration.

$$\text{Loss} = \sum_{\texttt{factor}} \sum_{t} \text{Loss}(\texttt{factor} = \texttt{value}, \text{iteration} = t). \tag{6}$$

## B.3 Discretization of Intermediate States

The training procedure described above applies to the *Continuous Latent Space Supervision*, *Discrete Latent Space Supervision*, and *Discrete Latent Space Supervision* ↻ methods in the same way. In the methods with a discrete latent bottleneck, we introduce an additional architectural mechanism where the read-out layers are used not only for computing the loss on the intermediate iterations, but also for mapping the latent representation to a discrete space.

In particular, letting $E_i^{(t)}$ be the embedding of the $i$-th token after $t$ recurrent iterations, we use argmax decoding of the linear read-outs to map the embedding to a discrete prediction for each factor. This discrete state is then re-embedded using the same learned embedding module to form the vectorized input $E_i^{(t+1)}$ at the next iteration. In particular, at iteration $t$, the model's forward pass is defined as follows

$$
\begin{aligned}
(\tilde{E}_1^{(t+1)}, \ldots, \tilde{E}_n^{(t+1)}) &\leftarrow \text{RecurrentTransformerBlock}(E_1^{(t)}, \ldots, E_n^{(t)}) \\
z_{i,\texttt{factor}}^{(t+1)} &\leftarrow \arg\max\{W_{\texttt{factor}} \tilde{E}_i^{(t+1)}\} \quad \texttt{factor} \in \{\texttt{syntax}, \texttt{variable}, \texttt{operation}, \texttt{value}\} \\
E_{i,\texttt{factor}}^{(t+1)} &\leftarrow \text{FactorEmbed}(z_{i,\texttt{factor}}^{(t+1)}) \quad \texttt{factor} \in \{\texttt{syntax}, \texttt{variable}, \texttt{operation}, \texttt{value}\} \\
E_i^{(t+1)} &\leftarrow E_{i,\texttt{syntax}}^{(t+1)} + E_{i,\texttt{variable}}^{(t+1)} + E_{i,\texttt{operation}}^{(t+1)} + E_{i,\texttt{value}}^{(t+1)}.
\end{aligned}
\tag{7}
$$

This discretization enables us to train the model with a type of *teacher-forcing* across recurrent iterations. That is, we can teacher-force the inputs $z_i^{(t)}$ at each iteration $t$. This enables more efficient training.

## B.4 Self-Correction Mechanism

In a reasoning task, each reasoning step depends crucially on the prior steps in the reasoning path. If a mistake is made at any stage, all subsequent computation is affected, and the error is often fatal. As the size of the problem and the number of computational steps scale, the likelihood of an error occurring at *some point* in the reasoning process becomes large, limiting the ability to generalize indefinitely to more complex problems. To address this challenge, a reasoning model must be able to detect and correct errors as they occur in order to recover when a mistake is made in its previous computation.

We train the model to detect and correct errors by randomly corrupting the model's latent state. That is, at each iteration, with some small probability, we corrupt a random selection of the value components of the models' discrete states. To achieve good loss, the model must learn to detect when a previously-computed value is incorrect and correct it before proceeding.

## B.5 Experiment Details & Additional Results

As with the baselines, we explore the effect of different architectural hyperparameters, such as positional encoding and the depth of the recurrent block, on model performance. Figure 11 depicts the average OOD performance as measured by the "% Fully Solved" metric for each model configuration in the *Discrete Latent Space Supervision*, and *Discrete Latent Space Supervision* ↻ methods. The results in the main text correspond to the best-performing models according to this metric. In particular, the best-performing *Discrete Latent Space Supervision* model is DeBERTa-L2H16D256, and the best-performing *Discrete Latent Space Supervision* ↻ model is DeBERTa-L4H16D384.

Figure 12 depicts additional experimental results for the *Discrete Latent Space Supervision*, and *Discrete Latent Space Supervision* ↻ methods. We highlight a few observations here:

- The positional encoding method is critical for length generalization. The DeBERTa positional encoding method (a relative positional encoding method) performed the best by far.

- 2 layers for the recurrent block were sufficient for the *Discrete Latent Space Supervision* method. However, the recorrection mechanism of *Discrete Latent Space Supervision* ↻ required a deeper recurrent block. We saw no significant improvement for the re-correction mechanism with 2 layers, but with 4 layers, the re-correction mechanism kicked in and enabled near-perfect OOD generalization.

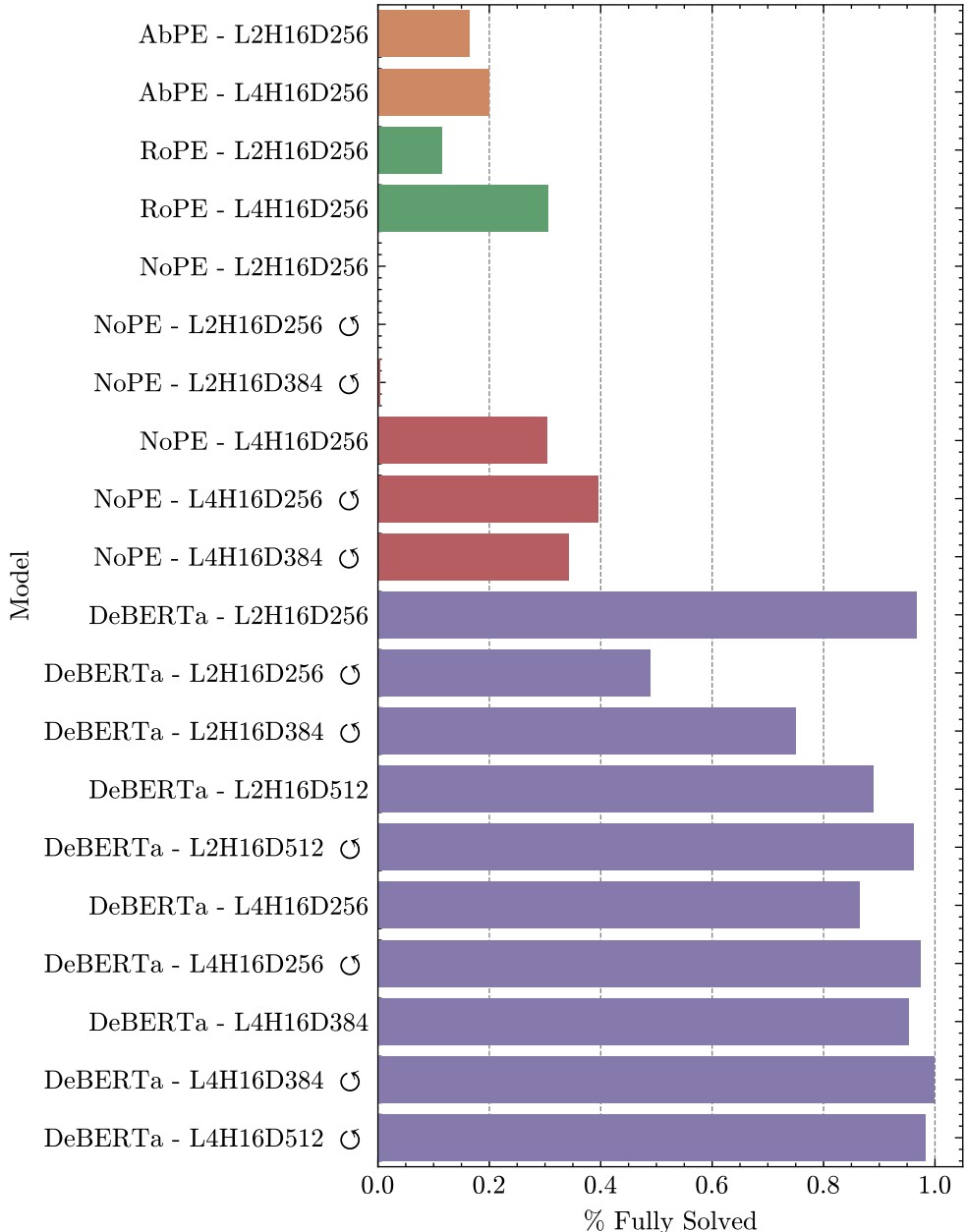

Figure 11: Average "% Fully Solved" across # nodes between 8 and 128, with training on ≤ 32 nodes.

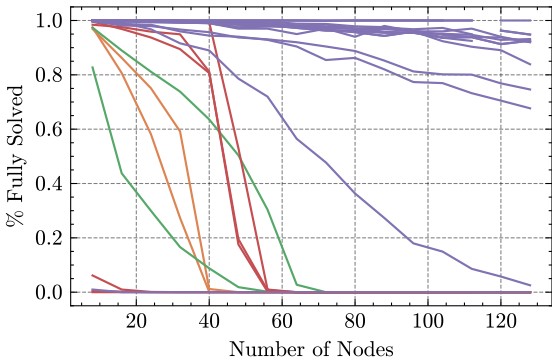

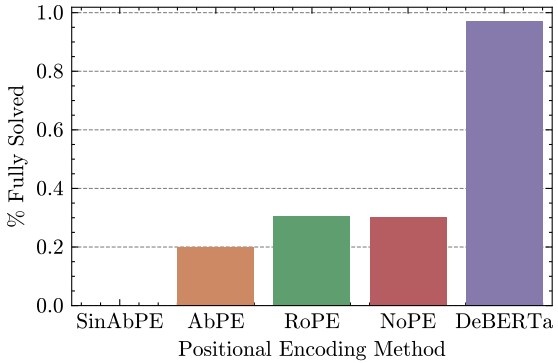

(a) Each line corresponds to an experimental run. Lines are color-coded by positional encoding, but other architectural hyperparameters vary and are not represented.

(b) Average OOD performance across test splits for the best model of each positional encoding method.

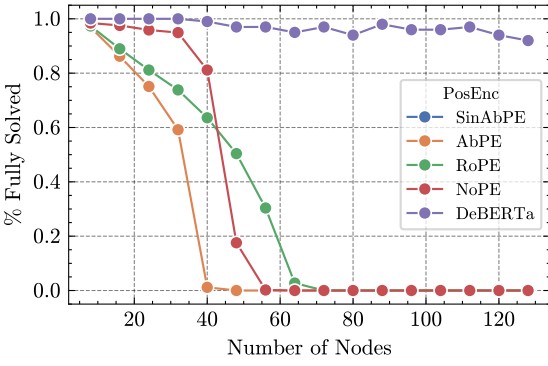

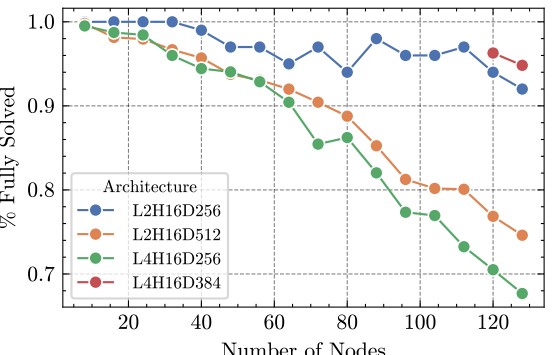

(c) % Fully solved by graph size for best model of each positional encoding method.

(d) % Fully solved by graph size for best model of each architectural configuration.

Figure 12: Further experimental results for methods exploring our proposed architectural mechanisms.

## C   Learning to infer an effective depth structure without any oracle decomposition

In the main experiments, the algorithm-alignment loss leverages oracle-provided depth information to expose a coarse layer-by-layer decomposition of the computation graph. This auxiliary supervision biases the model toward learning an iterative algorithm rather than a direct input–output mapping. While such supervision is widely available for synthetic algorithmic tasks—and is strictly weaker than the step-level traces required by token-space CoT training—it is natural to ask whether the model can *infer* the effective depth structure without any oracle decomposition.

This section presents a simple modification that removes the need for depth-based masking while preserving the benefits of iterative latent computation.

### C.1   Method: Iterative Refinement with ⟨EMPTY⟩-Discounted Loss

The key idea is to encourage the model to identify, during each recurrent iteration, which portions of the computation can already be solved and which should remain unsolved until later iterations. To make this possible without supervision on intermediate states, we augment the latent and prediction spaces with a special ⟨EMPTY⟩ token representing a "not yet solved" value, as in the model in the main paper.

At each recurrent iteration, the model predicts a distribution

$$p = \text{softmax}(\text{logits})$$

over all possible entries, including ⟨EMPTY⟩. Recall that in the main model, we apply a different loss for each iteration to align the learned algorithm with the input graph's depth structure. Here, we instead simply apply the same final-label loss at every iteration, but with a modified loss function.

For target label $y$ and discount factor $\alpha \in (0, 1)$, we use the discounted loss:

$$\mathcal{L}(p, y) = CE(p, y) \left(1 - \alpha \cdot p[\text{⟨EMPTY⟩}]\right), \tag{8}$$

where $CE(p, y)$ is the standard cross-entropy. The loss behaves as follows:

- If the model assigns probability mass to an *incorrect non-⟨EMPTY⟩* value, the penalty remains the full cross-entropy loss.

- If the model assigns probability mass to ⟨EMPTY⟩, the loss is discounted by a factor proportional to $\alpha$.

This creates a "safe" prediction state for positions the model is not yet ready to solve: predicting ⟨EMPTY⟩ is preferable to guessing an incorrect value, but still penalized so that the model eventually resolves all nodes. Over iterations, the recurrent update learns to fill in increasingly many positions as information accumulates in the latent state. Crucially, no oracle decomposition or depth mask is required.

As in the main experiments, we train with a fixed number of recurrent steps during training and allow the model to run until convergence at evaluation time.

### C.2   Results

Here, we present preliminary results with this method. The model achieves strong in-distribution performance and exhibits robust out-of-distribution generalization, approaching the performance of the full model that uses oracle depth supervision.

Qualitatively, the learned latent dynamics mirror those observed under oracle decomposition: early iterations populate easily solvable nodes, while later iterations resolve deeper nodes and correct earlier errors. Quantitatively, the models show high accuracy at depths far beyond the training regime, confirming that the latent recurrent architecture is capable of discovering the effective computational schedule without explicit supervision.

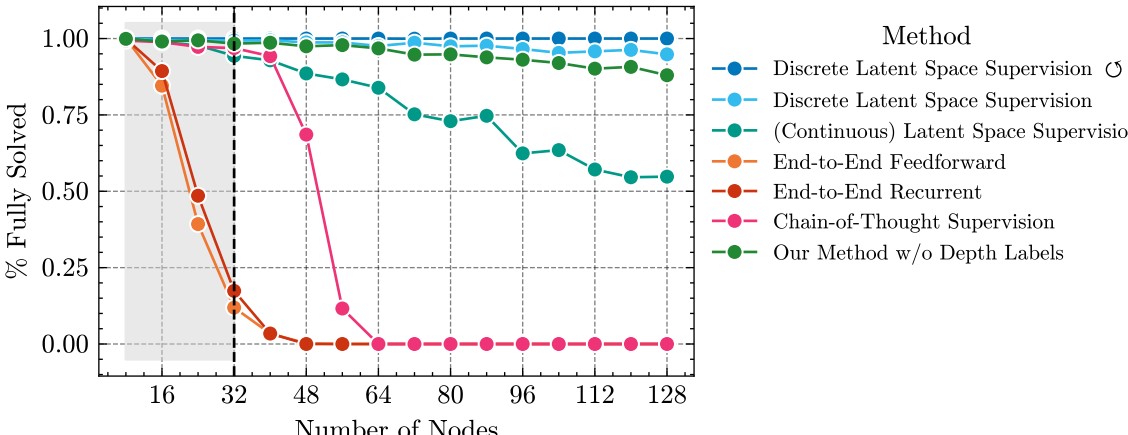

Figure 13: Our method achieves strong out-of-distribution generalization performance even without access to a depth decomposition oracle during training. Performance far exceeds the end-to-end and CoT baselines, and approaches the full method described in the main paper.

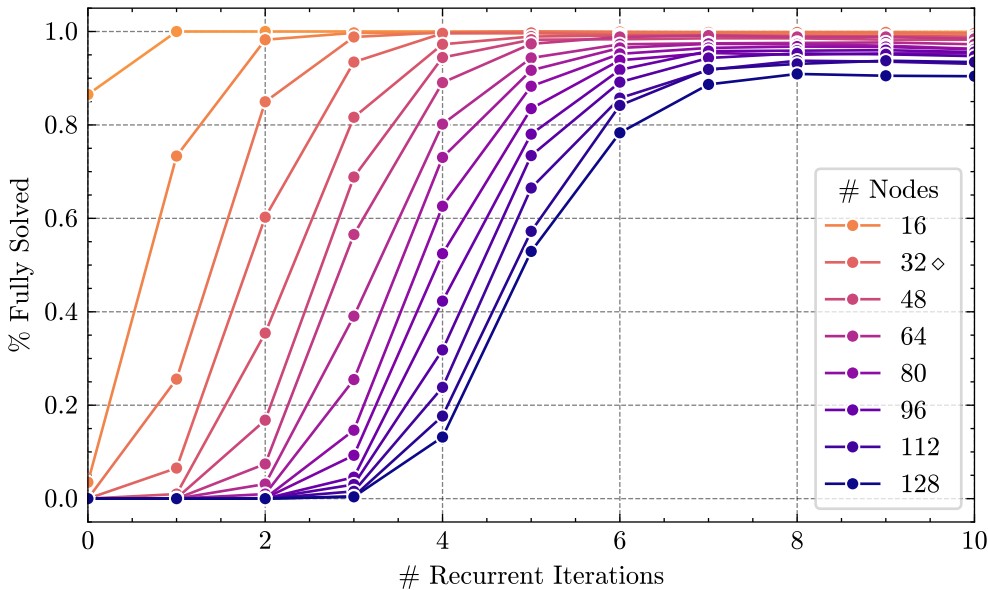

Figure 14: Proportion of input instances that are fully solved by number of iterations for graphs of varying sizes.

These findings indicate that explicit depth supervision is not required for the architecture to learn a depth-invariant algorithm. The ⟨EMPTY⟩-discounted loss provides a simple and effective way for the model to learn an iterative refinement strategy, broadening the applicability of the approach to domains where intermediate states are unavailable or costly to obtain.

# D   Details of Mechanistic Interpretability Analysis

In this section, we provide additional experimental evidence to support our claim on the mechanism learned by the model together with the error analysis of the model's predictions.

**Notice:**   The following analysis is conducted only to show the computation happening at the Right-Hand Side (RHS) position in each equation, as it is the place where the model is expected to compute the final result.

**Model Configuration.**   We use DeBERTa-L2H16D256 trained with our proposed *Discrete Latent Space Supervision* method (without the re-correction mechanism) on the mathematical reasoning task. Specifically, the recurrent Transformer model is configured with two layers, 16 attention heads, and a hidden dimension of 256. We use DeBERTa's relative positional encoding method. The training data is the same as that used in the main text. We choose this model setup because it is the best-performing configuration according to the "% Fully Solved" metric displayed in Figure 11 for a two-layer model. In particular, we cherry-pick the best-performing model trained with the same configuration with different random seeds, which has a "% Fully Solved" score of 99.98% on the OOD test set. We use this model to conduct the mechanism analysis for better interpretability. We train on the modular-23 addition task with a maximum graph size of 32. The total number of variables in the training data is 128. The testing data used for mechanism analysis has the maximum graph size of 128.

**Additional Definitions and Notations.**   In the following, we frequently use the following definitions and notations:

- **Head output**: For a given attention head $h$, we define the head output for a query vector $q_h \in \mathbb{R}^{d_h}$ for head dimension $d_h$ as

$$\text{Head Output}(h) = \text{softmax}(q_h K_h^\top / \sqrt{d_k}) V_h W_O^{(h)},$$

  where $K_h$ and $V_h$ are the key and value matrices of the head $h$, respectively, and $W_O^{(h)}$ is the output projection matrix of the head $h$. In the standard attention mechanism, each head's query, key, and value vectors are obtained by applying a linear transformation to the attention input specified by $W_Q^{(h)}$, $W_K^{(h)}$, and $W_V^{(h)}$, respectively. The above definition can be applied to define the head output for *any* query position. However, since our mechanism analysis focuses exclusively on the RHS position, we consistently define the head output as the output of the attention head at the RHS position. Here, we don't include the bias of the head output projection in the definition of the head output, as the bias applied to the final attention output is not specified to individual heads.

- **OV combined matrix**: For a group of attention heads $\mathcal{H} \subseteq [16]$, we define the OV combined matrix as

$$W_{OV}^{(\mathcal{H})} = \sum_{h \in \mathcal{H}} W_V^{(h)} W_O^{(h)},$$

  where $W_O^{(h)} \in \mathbb{R}^{d_h \times d}$ and $W_V^{(h)} \in \mathbb{R}^{d \times d_h}$ are the output projection matrix and the value projection matrix of the attention head $h \in \mathcal{H}$, respectively.

## D.1   First Layer Attention: Variable Copying

In the following, we will give a detailed analysis of the first layer attention mechanism.

### D.1.1   Group Structure in the First Layer Attention

**Group Structure in the First Layer Attention.**   First-layer attention heads exhibit a clear grouping pattern by the variable position they attend to. To rigorously demonstrate this grouping structure, we

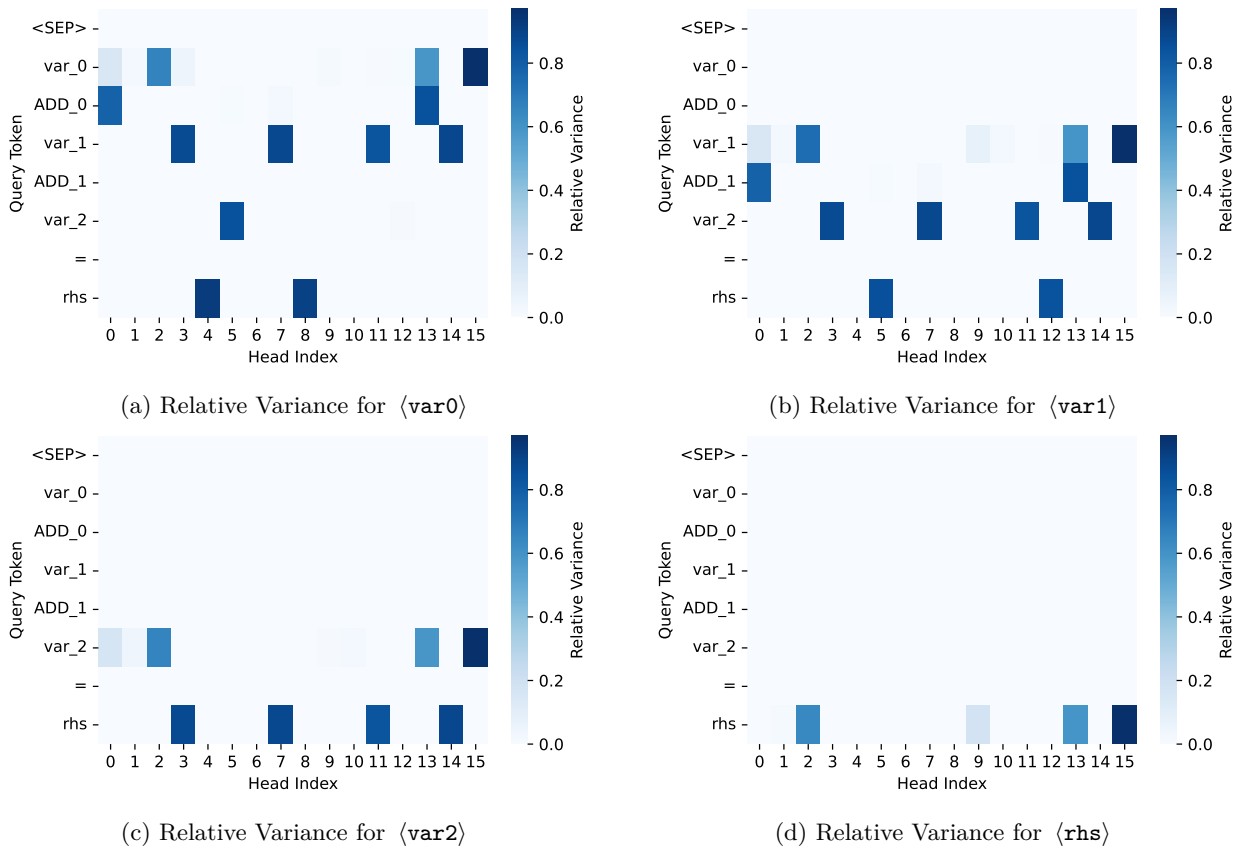

(a) Relative Variance for ⟨var0⟩        (b) Relative Variance for ⟨var1⟩

(c) Relative Variance for ⟨var2⟩        (d) Relative Variance for ⟨rhs⟩

Figure 15: Relative-variance heatmaps as we vary the value of ⟨var0⟩, ⟨var1⟩, ⟨var2⟩, and ⟨rhs⟩. Each row corresponds to a query position and each column corresponds to an attention head.

conduct controlled experiments by systematically varying the input data. On a test example with 128 nodes, we append a new **probe equation** to the end of the sequence with the following format:

$$[\texttt{sep}] \ \langle\texttt{var0}\rangle \ \langle+\rangle \ \langle\texttt{var1}\rangle \ \langle+\rangle \ \langle\texttt{var2}\rangle \ \langle=\rangle \ \langle\texttt{rhs}\rangle. \tag{9}$$

To identify the group structure in the first-layer attention and detect which heads belong to which groups, we measure each head's **relative variance** when we vary the value of each of the four variables ⟨var0⟩, ⟨var1⟩, ⟨var2⟩, and ⟨rhs⟩. See below for more details.

**Experiment Design for Group Structure Detection.** To detect which heads attend to ⟨var0⟩, we fix ⟨var1⟩, ⟨var2⟩, and ⟨rhs⟩ while randomly sampling different variables ⟨$x_i$⟩ with $i = 1, \ldots, 128$ for ⟨var0⟩. Note that the variable ⟨$x_i$⟩ must be computed in the preceding equations. Otherwise, the model cannot compute the value of ⟨$x_i$⟩ and the probe equation is invalid. As our testing data has all variables computed in the preceding equations, we collect 128 samples that only differ in the value of ⟨var0⟩. We then compute the relative variance of each attention head's output at each position within the probe equation across the 128 samples. Note that relative variance is a measure of how much the head's output varies in response to changes in ⟨var0⟩, and we give the rigorous definition in the next paragraph. The analysis can also be conducted for the other variable positions, and the results are reported in Figure 15.

**Relative Variance Calculation.** Let us take $n$ different sequences, e.g., the 128 sequences in the above experiment design. We only consider one RHS position for the probe equation in each sequence. For a given

attention head $h$, we define the relative variance over the $n$ sequences as

$$\text{Relative Variance}(h) = \frac{\text{tr}(\text{Cov}(\text{Head Output}(h)))}{\mathbb{E}[\|\text{Head Output}(h)\|_2^2]}. \tag{10}$$

Here, the covariance matrix for a sequence of vectors $v_1, \ldots, v_n$ is defined as

$$\text{Cov}(v_1, \ldots, v_n) = \frac{1}{n} \sum_{i=1}^{n} (v_i - \bar{v})(v_i - \bar{v})^\top,$$

where $\bar{v}$ is the mean of the sequence over the $n$ sequences, and $\mathbb{E}[\cdot]$ is the empirical expectation over the $n$ sequences. Intuitively, the relative variance measures how much the head's output varies relative to its overall magnitude. *A higher relative variance indicates that the attention head's output has a larger variance relative to its overall magnitude.* Since we only change $\langle \texttt{var0} \rangle$ in the above example, a larger relative variance for a head means that the head's output is primarily influenced by $\langle \texttt{var0} \rangle$.

**Illustration of Figure 15.** In Figure 15, we plot the relative variance heatmaps for all 16 attention heads when we vary the variable names of $\langle \texttt{var0} \rangle$, $\langle \texttt{var1} \rangle$, $\langle \texttt{var2} \rangle$, and $\langle \texttt{rhs} \rangle$. Each column corresponds to a different attention head, and each row corresponds to a different query position. As our goal is to understand the mechanism at the $\langle \texttt{rhs} \rangle$ position, we focus on the last row corresponding to the $\langle \texttt{rhs} \rangle$ query position in the figures. Each subfigure plots the relative variance heatmap for altering one particular variable. A higher relative variance in one subfigure indicates that the attention head's output is more sensitive to changes in the corresponding variable. Based on these results, we observe a clear group structure in the first layer's attention heads: Heads 4 and 8's relative variance is high only when we change the value of $\langle \texttt{var0} \rangle$, while the relative variance of the other heads is low. This fact suggests that heads 4 and 8 attend primarily to $\langle \texttt{var0} \rangle$. Similarly, heads 5 and 12 attend primarily to $\langle \texttt{var1} \rangle$, and heads 3, 7, 11, and 14 attend primarily to $\langle \texttt{var2} \rangle$. The last subfigure plots the relative variance heatmap for the $\langle \texttt{rhs} \rangle$ position. We observe that heads 2, 9, 13, and 15 attend to the RHS position, and the remaining heads do not exhibit a distinct pattern according to the relative variance heatmap. Notice that the above head groups are all disjoint. This further indicates that each head is specialized for a specific variable position. This result is also backed up by the trace of the attention logits of the first layer attention heads as shown in Figure 5.

**Summary of the Group Structure.** We observe that the attention heads in the first layer exhibit a clear grouping pattern based on which variable position they attend to. Therefore, we know that the first layer attention must be copying something from the LHS variables to the RHS position. In the following, we will conduct further analysis to identify what information is being copied.

### D.1.2 First Layer Attention Copying the Variable Identity

Here, by saying "copying the variable identity", we mean that the attention head is copying the factored embedding of `variable` among the four factored embeddings {`syntax`, `variable`, `operation`, `value`}. In the previous experiment, we have identified that the first layer attention heads are grouped into four groups, each of which attends to a specific variable position. Now, we aim to identify which of the four factored embeddings is being copied by these groups.

**Norm Amplification Analysis.** To achieve our goal, we analyze the norm amplification for each type of factored embeddings when passed through the combined OV matrix of different head groups. We define the norm amplification for a matrix $W_{OV}$ on input $x$ as:

$$\text{Norm Amplification}(W_{OV}, x) = \frac{\|W_{OV} x\|_2}{\|x\|_2}. \tag{11}$$

Note that the above definition can be applied to any matrix $W_{OV}$ and input $x$ with the conformal dimensions. For our analysis, we will consider $W_{OV}$ as the combined OV matrix of the attention heads in a group. Specifically, let $\mathcal{H} \subseteq [16]$ be a group of attention heads, and let $W_O^{(h)}$ and $W_V^{(h)}$ be the output projection

matrix and the value projection matrix of the attention head $h \in \mathcal{H}$, respectively. The combined attention OV matrix for a group $\mathcal{H} \subseteq [16]$ is then defined as

$$W_{OV}^{(\mathcal{H})} = \sum_{h \in \mathcal{H}} W_V^{(h)} W_O^{(h)}.$$

If the OV matrix is responsible for copying the identity of the variable, we expect to see a large amplification for the "`variable`" factored embedding, and a small amplification for the other types of embeddings {`syntax`, `operation`, `value`}. With a slight abuse of notation, for each `factored embedding type` $\in$ {`syntax`, `variable`, `operation`, `value`}, we can define the norm amplification as

$$\text{Norm Amplification}(W_{OV}^{(\mathcal{H})}, \texttt{factored embedding type}) = \mathbb{E}_{x \in \texttt{factored embedding type}} \left[ \frac{\|W_{OV}^{(\mathcal{H})} x\|_2}{\|x\|_2} \right].$$

Here, $\mathbb{E}_{x \in \texttt{factored embedding type}}$ is the average over the set of all factored embeddings of the same type. For example, if we consider the "`variable`" factored embedding type, we have

$$\text{Norm Amplification}(W_{OV}^{(\mathcal{H})}, \texttt{variable}) = \mathbb{E}_{x \in \texttt{variable}} \left[ \frac{\|W_{OV}^{(\mathcal{H})} x\|_2}{\|x\|_2} \right],$$

where $x$ iterates over all the 128 factored embeddings of the type `variable`. The results in Figure 5 (right) are computed by averaging the norm amplification over all the factored embeddings within each "`factored embedding type`". In Figure 16, we further histogram each factored embedding's norm amplification for different "`factored embedding type`" while different groups are highlighted in different colors, which provides a more detailed view of the norm amplification across different groups.

**Comparing the Norm Amplification Across Different Groups.** It can be observed from Figure 16 that the amplification factor for the "`variable`" factored embedding is significantly larger than that of the other types of embeddings, confirming our hypothesis that the OV matrix is responsible for copying the `variable` factored embeddings of the variable to the RHS position. We also observe that the amplification factor for the "`syntax`" factored embedding is also relatively large, which is consistent with the fact that the model is copying the variable identity.

From Figure 16, we confirm that for the first layer, attention has larger norm amplification for the "`variable`" factored embedding ($\approx 15$) than the other types of embeddings ($\approx 5$). In particular, the larger norm from the "other" group as shown in Figure 5 is due to the self-copying operation of the attention head 15 at the RHS position.

**Additional Evidence on change of number of variables.** We provide one interesting observation on how the model handles different numbers of variables in the input equations in Figure 17. Head 4 and head 8 are the two attention heads that attend to the first variable position in the first layer attention when the number of variables is 3. When the number of variables is changed to 2, we observe that head 4 now attends to the [`sep`] token, while head 8 attends to the equal sign token of the previous equation. This indicates that the equal sign token and the [`sep`] token act as *attention sink* for head 4 and head 8, respectively.

### D.1.3 First Layer MLP Residual Stream Does Not Change the Residual Stream Significantly

We measure the changes brought by the first MLP layer to the residual stream by computing the Relative L2 Error as:

$$\text{L2 Relative Error} = \frac{\|\text{Residual Before MLP} - \text{Residual After MLP}\|_2}{\|\text{Residual Before MLP}\|_2}.$$

This metric quantifies how much the MLP alters the original residual signal. Figure 18 illustrates the heatmap of the relative L2 error computed at the $\langle$`rhs`$\rangle$ position across a set of 256 samples. We observe that the relative L2 error is relatively small, which indicates that the MLP does not change the residual stream significantly.

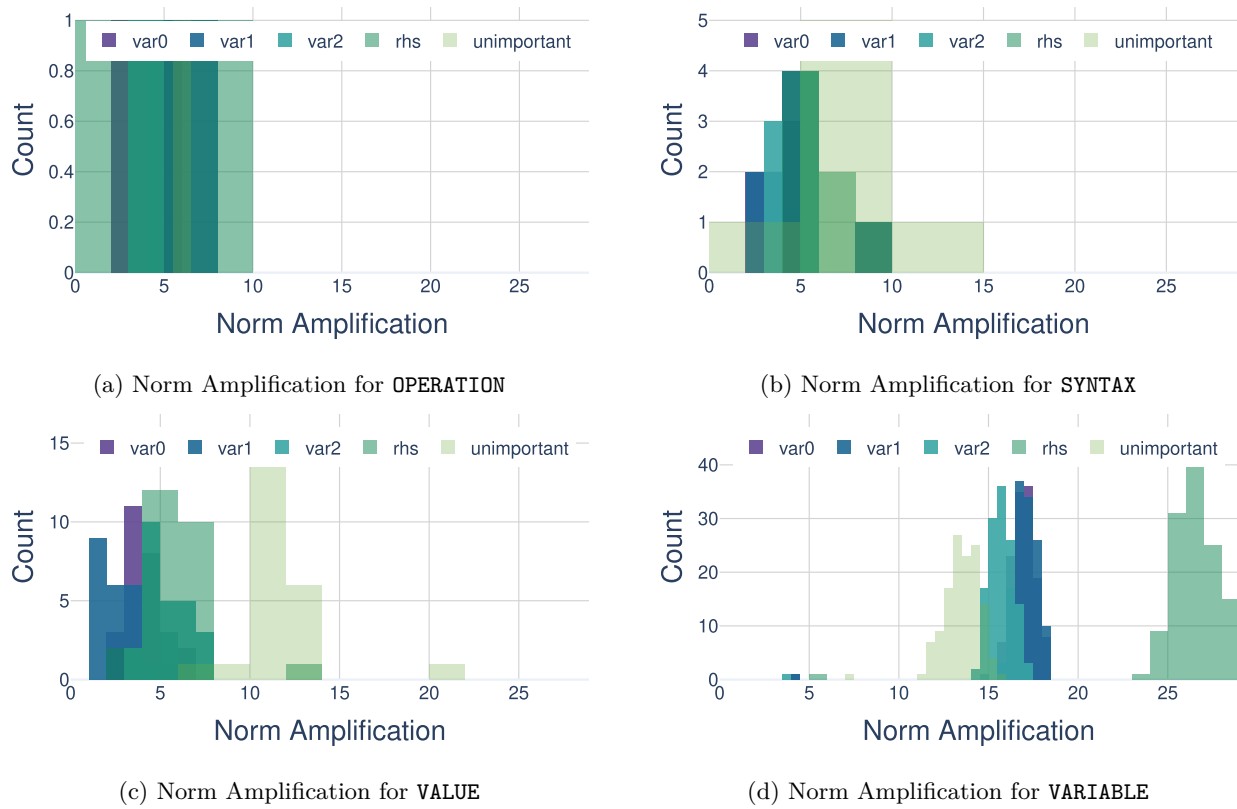

(a) Norm Amplification for `OPERATION`

(b) Norm Amplification for `SYNTAX`

(c) Norm Amplification for `VALUE`

(d) Norm Amplification for `VARIABLE`

Figure 16: Histogram of norm amplification (defined in (11)) for the embeddings in the four factored embedding types {`syntax`, `variable`, `operation`, `value`} when passed through the first attention layer's combined OV matrix. Each subfigure contains five histograms in different colors, while each histogram corresponds to a different group of attention heads' combined OV matrix. Here, the 16 attention heads are grouped by the different variables they attend to, which are ⟨var0⟩, ⟨var1⟩, ⟨var2⟩, and ⟨rhs⟩, and an additional group for the heads that do not demonstrate a clear pattern.

## D.2 Second Layer Attention: Value Copying

**A Hypothesis on the Second Layer Attention Heads.** As we have shown previously, the first layer attention heads copy the `variable` factored embeddings of the variables to the RHS position, which tells the model the identity of all the variables on the LHS of an equation. To compute the final answer for the RHS position, the model still needs to copy the values of the variables ⟨var0⟩, ⟨var1⟩, and ⟨var2⟩ to the RHS position. Thus, we hypothesize that the second layer attention heads will copy the values of the variables ⟨var0⟩, ⟨var1⟩, and ⟨var2⟩ to the RHS position.

**The Second Layer Attention Heads also Have a Group Structure** To test this hypothesis, we prepare data that contains probe equations of the same form in (9) and conduct a controlled experiment designed to analyze how attention heads respond to changes in individual variable values. Different from the previous experiment where we change the variable identity, this time we fix the variable identity and only change the value of each variable ⟨var0⟩, ⟨var1⟩, and ⟨var2⟩ one at a time while keeping the other two variable values fixed. This is achieved by altering the previous equations that compute the value of the variable to be changed. Specifically, for each of the three variables ⟨var0⟩, ⟨var1⟩, and ⟨var2⟩, we conduct a separate experiment where we collect $N$ samples by varying only that variable's value while keeping the other two variables fixed. Then, for each variable ⟨var i⟩, we collect the second layer attention head outputs across the $N$ samples at the RHS position of the probe equation, and compute the following metrics:

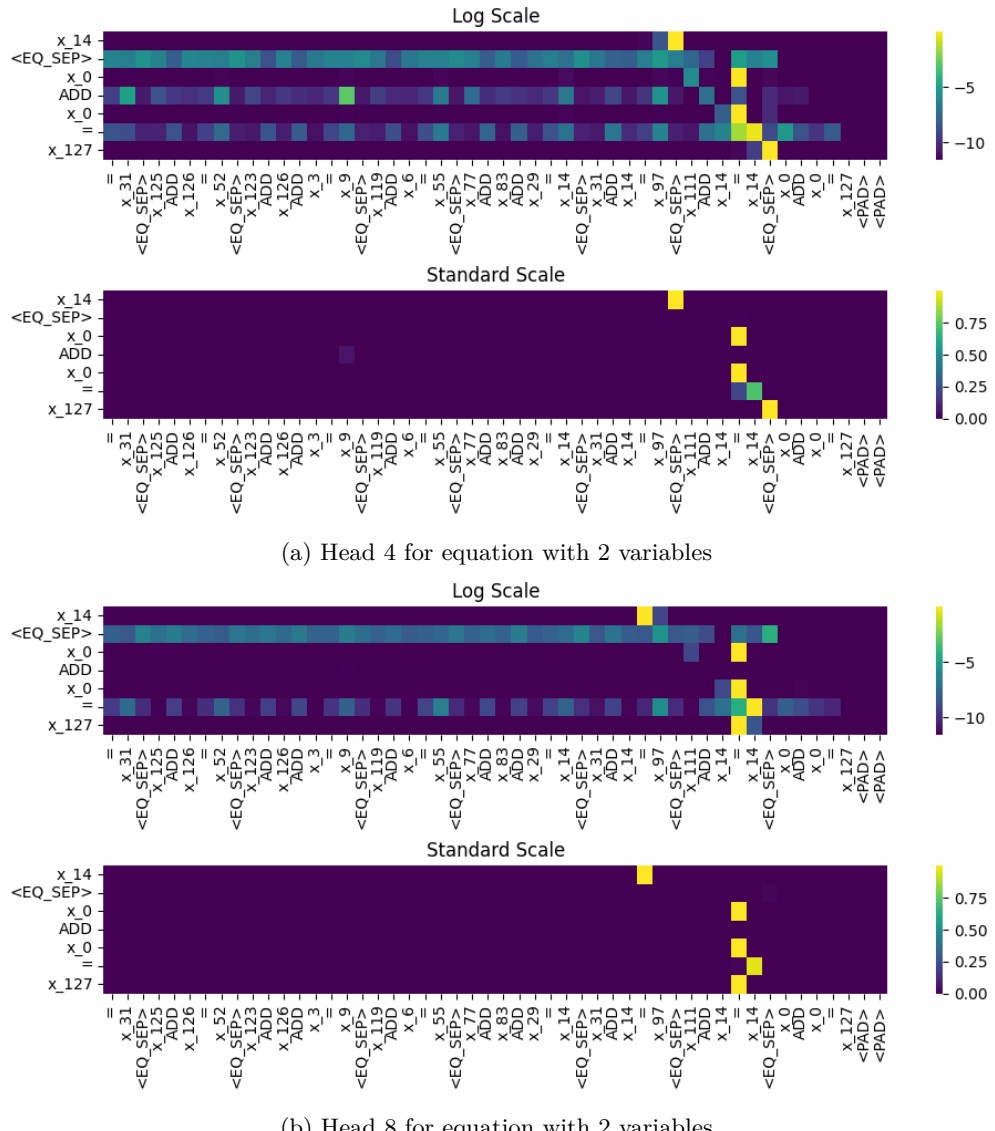

(a) Head 4 for equation with 2 variables

(b) Head 8 for equation with 2 variables

Figure 17: Visualization of the attention maps for the head group that attends to the first variable position in the first layer attention, which includes head 4 and head 8. Each row corresponds to a different query position, and each column corresponds to a different key position. We only show the rows within the last probe equation, and the columns within the last 50 positions in the sequence. Here, we notice that at the RHS query position (token $\langle x_{127} \rangle$ in the last row), head 4 attends to the [sep] token and attention head 8 attends to the equal sign token

- The variance of the outputs (numerator in (10))

- The average squared norm (denominator in (10))

- The relative variance (ratio of the above quantities)

These metrics help us identify which heads are sensitive to changes in each variable's value. The results are shown in Figure 19. We deduce from the results (especially the relative variance) that the Heads (0, 8, 15) form the first group, which copy the value for $\langle \text{var0} \rangle$; Heads (5, 10) form the second group, which copy the value for $\langle \text{var1} \rangle$; and Heads (2, 3, 4, 7, 9) form the third group, which copy the value for $\langle \text{var2} \rangle$.

L2 Relative Error Distribution

Figure 18: Histogram of the L2 relative error between the residual stream before and after the first layer MLP.

**Second Layer Attention Heads Copy the Values of the Variables to the RHS Position.** Similar to the experiment in the first layer, we compute the norm amplification coefficient for the second layer attention heads' OV matrix, combined by groups, as shown in Figure 20. We observe that the norm amplification coefficient for the `value` factored embedding is significantly larger than that of the other types of embeddings, confirming our hypothesis that the OV matrix is responsible for copying the `value` factored embeddings of the variable to the RHS position.

### D.3 Second Layer MLP: Modular Addition in the Frequency Domain

**Extracting the Copied `value` Factored Embeddings.** After confirming that the second layer attention heads copy the values of $\langle$var0$\rangle$, $\langle$var1$\rangle$, and $\langle$var2$\rangle$ to the RHS position, we now look closer at how the `value` factored embeddings of all three variables coexists in the residual stream after the second layer attention. To do so, we need to determine what should be the "`value`" factored embedding after passing through the second layer attention. Note that in the second layer attention, the copied `value` factored embedding for each $\langle$var i$\rangle$ are passed through the OV combined weight matrix for the corresponding group of attention heads, where the group structure is already determined in the previous experiment. For this reason, we can define the new "`value`" factored embedding at the output of the second layer attention as:

$$\text{new value}(i) = W_{OV}^{(\mathcal{H}_i)} \cdot \text{value}(i),$$

where $\text{new value}(i)$ represents the new "`value`" factored embedding for $\langle$var i$\rangle$, and $\mathcal{H}_i$ represents the group of attention heads that copy the value of $\langle$var i$\rangle$ for $i = 0, 1, 2$. We also consider the same definition for the embedding of "N/A" and "`empty`" in the value factor. Therefore, we have in total 75 new "`value`" factored embeddings, where the first 25 are for $\langle$var0$\rangle$, the next 25 are for $\langle$var1$\rangle$, and the last 25 are for $\langle$var2$\rangle$. We plot the cosine similarity among the new "`value`" factored embeddings as shown in Figure 21.

From Figure 21, we can observe two interesting phenomena:

- The `value` factored embeddings for the three variables are almost orthogonal for two different variables, but not for the embeddings within the same variable.

- The `value` factored embeddings for the same variable show a periodic pattern.

The periodic pattern implies that the `value` factored embeddings for the same variable are likely formed by some combination of sin and cos functions. Therefore, it will be easier to understand the modular addition operation in the frequency domain.

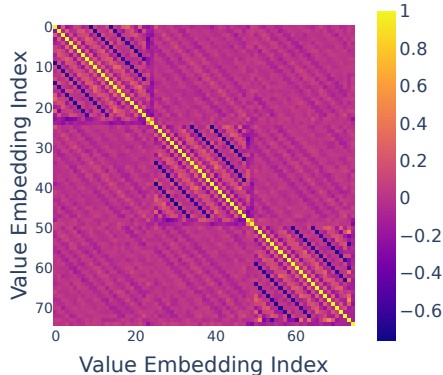

Figure 21: Cosine similarity of the **new value** factored embeddings for all three variables in the residual stream after the second layer attention.

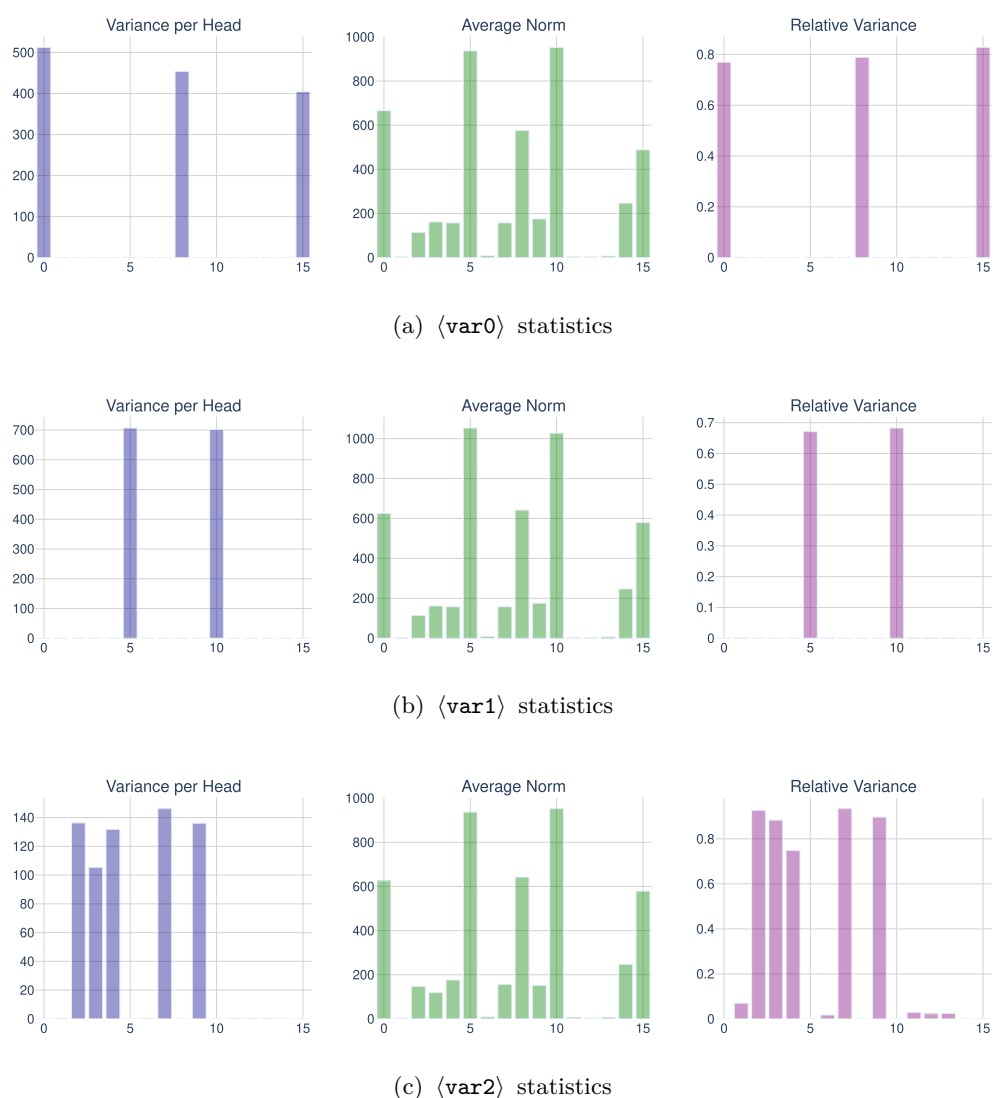

Figure 19: Attention head statistics for the second layer attention. Each subfigure shows three histograms corresponding to the variance (numerator to (10)), average norm (denominator to (10)), and relative variance for each attention head's outputs.

**Modular Addition in the Frequency Domain.** To systematically analyze how the model performs the modular addition operation, we prepare an equation of the form in (9), and we change the previous equations to alter the value of each variable $\langle \texttt{var0} \rangle$, $\langle \texttt{var1} \rangle$, and $\langle \texttt{var2} \rangle$. Specifically, we let $\langle \texttt{var0} \rangle$, $\langle \texttt{var1} \rangle$, and $\langle \texttt{var2} \rangle$ iterate over the set $\{0, 1, 2, \ldots, 22\}$ since the model is trained on modular-23 addition. To study how the MLP performs the modular addition operation, we pick the following four positions in the model: (i) pre-activation of the second layer's MLP, (ii) post-activation of the second layer's MLP, (iii) the output of the second layer's MLP, and (iv) the model's decoder output. For each of these positions, we take the vector obtained at the RHS position of the prepared equation, where we denote such vector as $v(a, b, c)$ with dimension $d$ when the input variables are $\langle \texttt{var0} \rangle = a$, $\langle \texttt{var1} \rangle = b$, and $\langle \texttt{var2} \rangle = c$. We then compute the 3-dimensional 23-point Discrete Fourier Transform (DFT) applied independently to

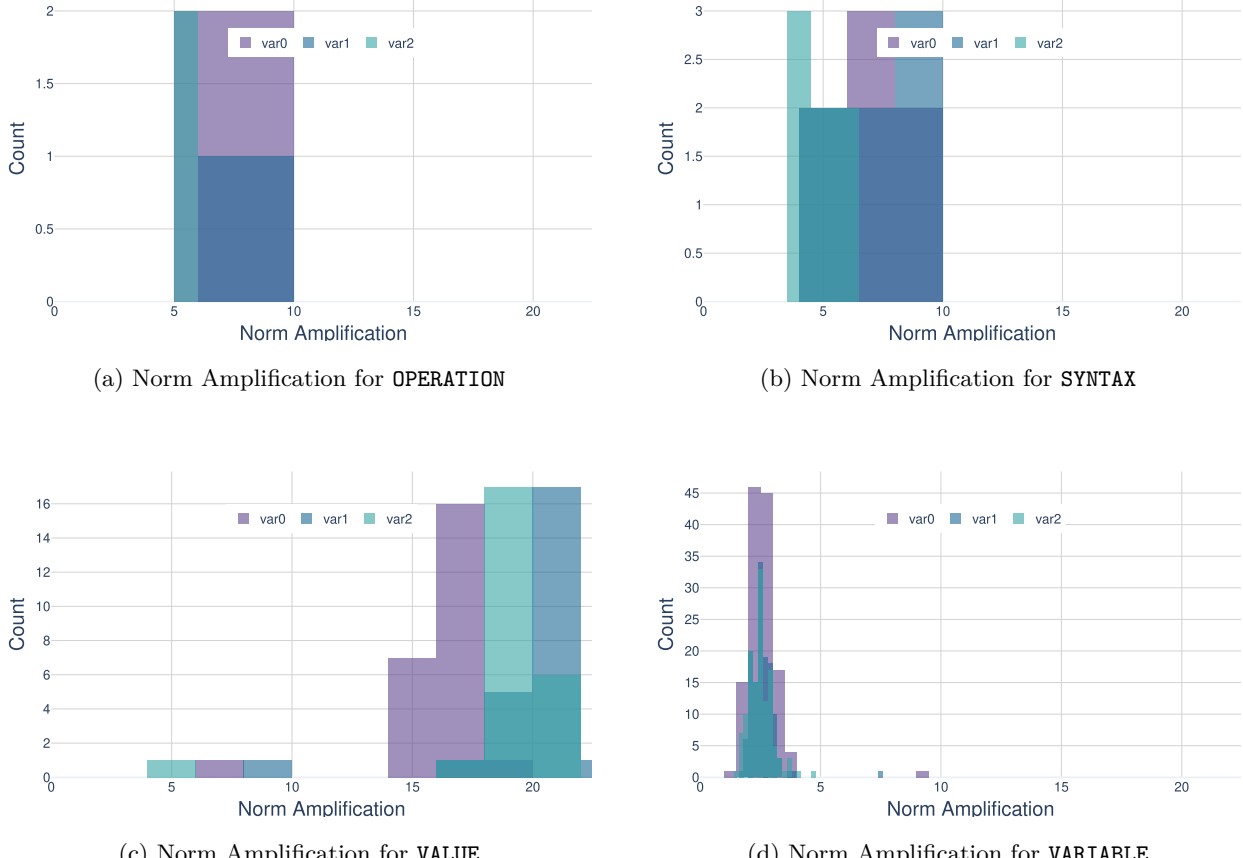

(a) Norm Amplification for `OPERATION`

(b) Norm Amplification for `SYNTAX`

(c) Norm Amplification for `VALUE`

(d) Norm Amplification for `VARIABLE`

Figure 20: Histograms of norm amplification for the four factored embedding types in the second layer attention's OV matrix. The 16 attention heads are grouped by the variable they attend to, which are $\langle \text{var0} \rangle$, $\langle \text{var1} \rangle$, and $\langle \text{var2} \rangle$. In each subfigure, we make three histograms each corresponding to the combined OV matrix for each group of attention heads. The three histograms in each subfigure are shown in different colors, and each histogram is for all the embeddings of the corresponding factored embedding type. It can be observed that the amplification factor for the "`value`" factored embedding is significantly larger than that of the other types of embeddings, confirming our hypothesis that the OV matrix is responsible for copying the "`value`" factored embeddings of the variable to the RHS position.

each coordinate of $v$ over $(a, b, c)$, which is defined as:

$$\text{DFT}_3(v)_{j,k,l} = \frac{1}{\sqrt{23^3}} \sum_{a=0}^{22} \sum_{b=0}^{22} \sum_{c=0}^{22} v(a, b, c)\, e^{-2\pi i \frac{aj+bk+cl}{23}}, \quad j, k, l = 0, 1, \dots, 22,$$

The obtained DFT tensor is a 4D tensor with dimension $23^3 \times d$. We then compute the norm of the DFT tensor along the last dimension, which represents the magnitude of the corresponding frequency component. Since the obtained DFT tensor is conjugate symmetric, we have

$$\text{DFT}_3(v)_{j,k,l} = \overline{\text{DFT}_3(v)_{22-j,22-k,22-l}},$$

Therefore, we only need to focus on the first half of the tensor, which has dimension $12^3$.

**Studying the DFT Tensor by Frequency Group.** We further partition the tensor into 7 groups by the algebraic patterns of the frequency component $(j, k, l)$:

- Group 1: $(0, 0, 0)$

- Group 2: $(0, 0, a)$, $(0, a, 0)$, $(a, 0, 0)$ for $a \neq 0$

- Group 3: $(0, a, b)$, $(a, 0, b)$, $(a, b, 0)$ for nonzero $a \neq b$

- Group 4: $(0, a, a)$, $(a, 0, a)$, $(a, a, 0)$ for $a \neq 0$

- Group 5: $(a, b, c)$ for nonzero $a \neq b$, $b \neq c$, $c \neq a$

- Group 6: $(a, a, b)$, $(a, b, a)$, $(b, a, a)$ for nonzero $a \neq b$

- Group 7: $(a, a, a)$ for $a \neq 0$

We then histogram the norm of the DFT tensor in the last dimension for each group, as shown in Figure 22. At the pre-activation stage of the second layer MLP, the DFT tensor shows its highest norm for the group $(0, 0, 0)$, which suggests a dominant bias term that is independent of the input variables. Progressing from the pre-activation (Figure 22a) to the MLP output (Figure 22c), this bias term gradually diminishes, while the norm corresponding to the group $(a, a, a)$ steadily increases. This trend indicates that the MLP output contains a strong frequency component of the form

$$\cos\left(\frac{2\pi a x}{23}\right) \cdot \cos\left(\frac{2\pi a y}{23}\right) \cdot \cos\left(\frac{2\pi a z}{23}\right), \tag{12}$$

or a similar combination involving both sine and cosine functions with the same frequency $a$. In (12), $x$, $y$, and $z$ denote the value of the three variables in the equation, and $a$ is the frequency. The term in (12) corresponds to a degree-3 term on frequency $a$, indicating that the model is capable of computing terms in the form of $\cos(2\pi a(x + y + z)/23 + \varphi)$ for some frequencies $a$ and phase $\varphi$, and eventually decodes to the correct answer $x + y + z \mod 23$.

### D.4 Error Analysis

To better understand the probed model's performance, we analyze its prediction errors. As we have three functional components in the model—the first layer attention, the second layer attention, and the last feedforward layer—we consider three sources of errors: *(i)* the first layer's attention mapping copies from the wrong variable position, *(ii)* the second layer's attention fails to copy the correct variable value, and *(iii)* the feedforward layer miscalculates the sum of the LHS variables. An account of the errors by source is shown in Table 2, where the major source of error is the feedforward layer calculation. Note that when considering the three sources of errors, if the error *(i)* occurs, we don't count towards error *(ii)* and *(iii)*. Similarly, when error *(ii)* occurs, we don't count towards error *(iii)*. In the following, we details how we identify the three sources of errors.

*May need to update fig x-axis to make more clear. – Awni*

#### D.4.1 Identifying Different Sources of Errors

When our loop transformer model is computing the RHS value for all the equations in the sequence, we have two key concepts:

- **Depth of equation**: The depth of an equation is the number of iterations required to compute the correct RHS value. More formally, the depth of an equation is the depth of the RHS variable in the computation graph. Take Figure 1 as an example, the depth of the equation "$20 = x_7$" is 1, as the model only needs a single loop to compute the correct RHS value, and the depth of the equation "$x_7 + x_{42} = x_{23}$" is 2, as the model needs two loops to compute the correct RHS value.

- **Number of iterations**: The number of iterations describes how many times the loop transformer model has iterated over the input sequence.

By definition, the minimum number of iterations needed for computing the correct RHS value of an equation of depth $d$ is at least $d$. In fact, we observe that most of the equations can be computed with exactly the number of iterations equal to the depth. For this reason, we only consider the equations and the number of iterations such that

$$\texttt{depth of equation} \geq \texttt{number of iterations}, \quad \text{or for short,} \quad \texttt{depth} \geq \texttt{iter}. \qquad (13)$$

Moreover, we do not add any probe equations in this error analysis. This means that we apply the knowledge learned from the previous experiments with probe equations to identify errors happening in the whole sequence.

In the following, we details how we identify the three sources of errors.

**First Layer Attention Error.** We identify first layer attention errors by analyzing how well each attention head group focuses on its assigned variable position. For each equation's RHS position, we examine the attention map (an $H \times L \times L$ tensor, where $H$ is the number of attention heads and $L$ is the sequence length) to extract the relevant attention probabilities.

Consider a concrete example: For the head group $\mathcal{H}_0$ that is responsible for attending to $\langle\texttt{var0}\rangle$, we look at the attention probabilities where the query is at the $\langle\texttt{rhs}\rangle$ position and the key is at the $\langle\texttt{var0}\rangle$ position, for all heads in $\mathcal{H}_0$. We then average these probabilities within the head group.

Table 2: Attribution of errors by source in the testing dataset with $N = 128$ and 23k sentences.

| Error Source | Count |
|---|---|
| First Layer Attention Error | 9 |
| Second Layer Copy Error | 1 |
| Feedforward Calculation Error | 30 |
| **Total** | **40** |

For each equation, we can use the above strategy to obtain a single **group-wise attention probability** for each head group at the $\langle\texttt{rhs}\rangle$ position. If this group-wise attention probability is less than our threshold of 0.9, we classify it as a first layer attention error, indicating that the head group failed to maintain sufficient focus on its designated variable position. In fact, the error analysis is not very sensitive to the choice of the threshold. As we will see later in Figure 23 (Top Row), the computed group-wise attention probability is either very close to 1 or very close to 0 (for $\langle\texttt{var0}\rangle$ and $\langle\texttt{var2}\rangle$, where $\langle\texttt{var1}\rangle$ has a slightly larger deviation from 1 on the high end). It is very easy to identify when an error occurs in the first layer attention.

**Second Layer Copy Error.** For the second layer attention, we analyze the attention head's output rather than the attention map. This approach is necessary because the "`value`" factored embedding from the first layer may be distributed across multiple positions, including special tokens (like delimiters or operators), rather than being confined to the original variable position. Fortunately, we already have the extracted "`new value`($i$)" factored embeddings for each $\langle\texttt{var i}\rangle$ in the previous experiment. We thus treat these "`new value`($i$)" factored embeddings as the ground truth value embeddings for $\langle\texttt{var i}\rangle$ in the second layer attention output.

For each equation containing $\langle\texttt{var i}\rangle$, we compute the cosine similarity between the ground truth value embedding for $\langle\texttt{var i}\rangle$ and the designated head group's output at the $\langle\texttt{rhs}\rangle$ position in the second layer attention. We call this cosine similarity the "group-wise cosine similarity". If the cosine similarity is less than our pre-determined threshold of 0.9, we consider it a second layer copy error for that head group, indicating the model fails to copy the correct variable value to the RHS position.

Similar to the first layer attention analysis, the choice of threshold is not critical. As shown in Figure 23 (Middle Row), the cosine similarity between the second layer attention outputs and the target value embeddings exhibits a clear pattern: either very close to 1 for correct copies, or significantly lower for incorrect copies. This stark separation makes it straightforward to identify second layer copy errors.

**Feedforward Calculation Error.** The feedforward calculation error is defined in the following way: If an equation passes the first two error checks, meaning that the first layer attention successfully attends to the correct variable position, and the second layer attention successfully copies the correct variable value to the

RHS position, but the model still makes a mistake when applying the factored decoder after the second layer MLP, we consider it a feedforward calculation error.

An account of the errors by source is shown in Table 2, where the major source of error is the feedforward layer calculation. Overall, the model demonstrates remarkable accuracy, where the total number of errors is only 40 out of 23k examples. A more detailed analysis of the errors is shown in Figure 23.

### D.4.2 Additional Error Analysis

Here, we provide additional evidence for the above discussion. In Figure 23, instead of just counting the number of times a specific error occurs, we histogram all the statistics used by the above error analysis procedure. Figure 23 (Top Row) is a histogram of the group-wise attention probability in the first layer, organized by three head groups $\mathcal{H}_0$, $\mathcal{H}_1$, and $\mathcal{H}_2$, where each $\mathcal{H}_i$ is responsible for copying the value of $\langle \texttt{var i} \rangle$. See the "First Layer Attention Error" paragraph above for more details. We see that the attention scores generally concentrate their probability mass around 1 on the correct variable; however, the heads responsible for copying $\langle \texttt{var2} \rangle$ are somewhat less concentrated, resulting in more errors. Moreover, for some examples where the final prediction is incorrect, we observe a clear error pattern in the histogram: the attention head group completely fails to attend to the correct variable position, with the group-wise attention probability dropping to nearly 0. This stark contrast between successful and failed attention patterns makes it easy to identify first layer attention errors.

In addition, Figure 23 (Middle Row) shows histograms of the cosine similarity between the second layer attention outputs and the target value embedding, again for all three head groups. For most examples, the cosine similarity is close to 1, showing that the second layer retrieves the value embeddings. However, for some examples where the final prediction is incorrect, we also observe a clear error pattern in the histogram: the cosine similarity drops to nearly 0. This stark contrast between successful and failed second layer copy patterns makes it easy to identify second layer copy errors as well.

***Does the Model Perform Self-Correction?*** The first two rows in Figure 23 are reported only for equations with `depth` $\geq$ `iter`. This is because the number of iterations required for computing the correct RHS value of equations is at most its depth. However, if we let the number of iterations go beyond the depth of the equations, as shown in Figure 23 (Bottom Row), the first layer attention heads are not able to concentrate their probability mass on the correct variable. This finding indicates that there is no further computation performed by the model at an equation position after the number of iterations reaches the depth of the equations, hence the model does not perform self-correction. One possible explanation for this to happen is the use of weight-decay in the training process. As the value for the $\langle \texttt{rhs} \rangle$ variable is already computed after the number of iterations reaches the depth of the equations, the model can directly pass on the computed value to the next iteration via the residual stream without any further computation.

How to let the model perform self-correction? We observe that the model does not perform self-correction because we only train the model on "perfect" data, where the model has no need to perform any further computation beyond the depth of the equations. In fact, we can let the model perform self-correction by training the model on "imperfect" data, where the model has to perform some further computation beyond the depth of the equations. This motivates our proposal of *Discrete Latent Space Supervision* ↻ method, which trains the model with corrupted data to teach the model to recover from errors. Consequently, increasing the number of iterations beyond the depth of the input can be useful because it allows the model to correct any errors in previous iterations.

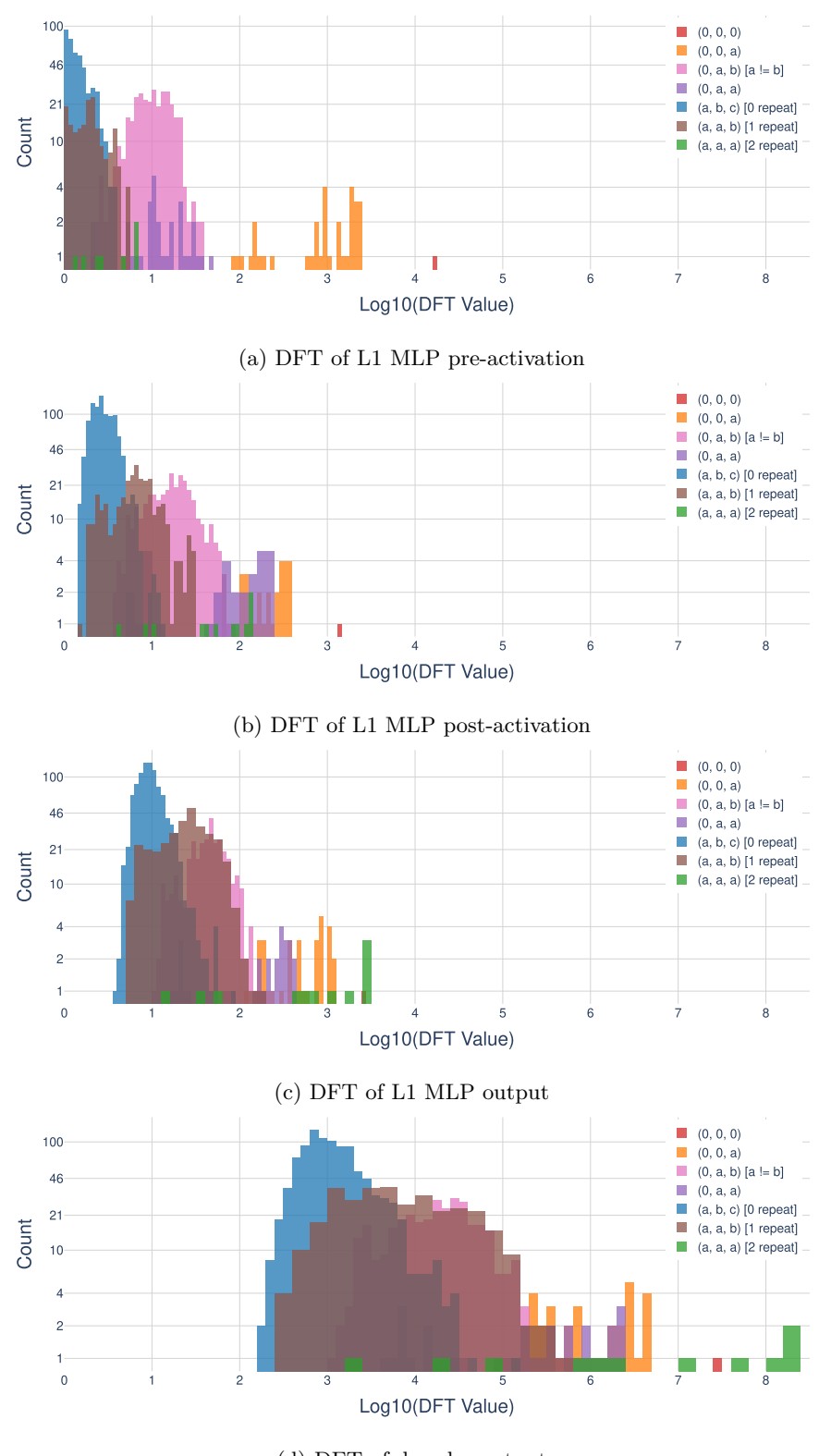

(a) DFT of L1 MLP pre-activation

(b) DFT of L1 MLP post-activation

(c) DFT of L1 MLP output

(d) DFT of decoder output

Figure 22: Combined DFT histograms for the second layer MLP pre-activation, MLP post-activation, MLP output, and decoder output.

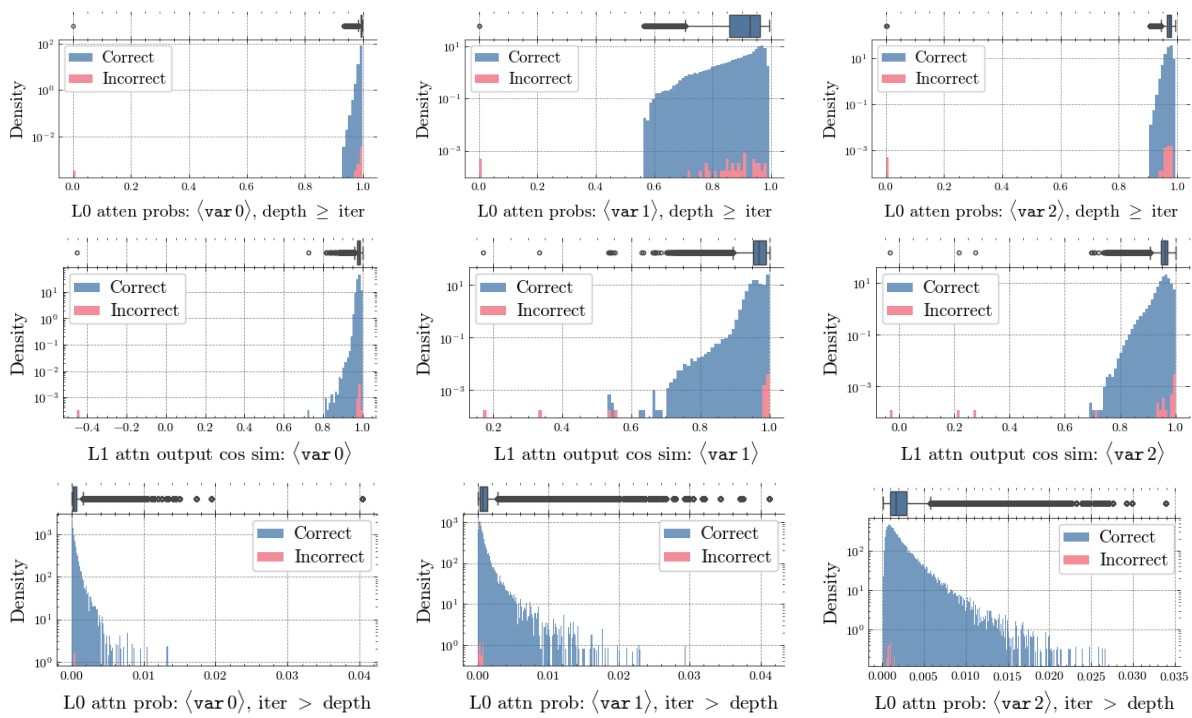

Figure 23: Error analysis. **Top Row.** Histograms for the group-wise attention probability in the first layer for all three head groups attending to $\langle\texttt{var0}\rangle$, $\langle\texttt{var1}\rangle$, and $\langle\texttt{var2}\rangle$, respectively. Here, the target equations considered all satisfy $\texttt{depth} \geq \texttt{iter}$ as defined in (13). We use different colors to separate the equations based on whether the decoded RHS value is correct or not after the second layer MLP. **Middle Row.** Histograms of the group-wise cosine similarity for the second layer attention head groups' outputs with the target values' embedding. Only equations with $\texttt{depth} \geq \texttt{iter}$ are included. **Bottom Row.** Histograms of the group-wise attention probability in the first layer for all three head groups. Here, the target equations considered all satisfy $\texttt{depth} < \texttt{iter}$, meaning that the number of iterations is beyond the depth of the equations.

