# OpenReview forum: "Unlocking Out-of-Distribution Generalization in Transformers via Latent Space Reasoning"
_TMLR — Under review for TMLR_

### Review · Reviewer_STVW · 2026-07-02

**Summary Of Contributions:**

This paper studies out-of-distribution (specifically length/depth) generalization in Transformers on a synthetic modular arithmetic circuit-evaluation task (mod 23, up to N=128 nodes, trained on N≤32). The authors propose combining four mechanisms atop a recurrent Transformer block: (1) input-adaptive recurrence, where the number of recurrent iterations scales with circuit depth; (2) latent-state algorithmic supervision, a depth-indexed intermediate loss supervising node values layer-by-layer in latent space; (3) a discrete bottleneck that argmax-decodes hidden states into a factored symbolic space (syntax/variable/operation/value) and re-embeds between iterations, yielding a fixed-point halting rule; and (4) a self-correction scheme that corrupts latent values during training so the model learns attractor-like repair dynamics.

Empirically, the full model ("Discrete Latent Space Supervision ⟲") reportedly achieves near-perfect accuracy across all OOD splits up to N=128, while End-to-End and CoT baselines collapse near N≈40. The paper's strongest component is its mechanistic interpretability analysis (Section 5, Appendix D), which reverse-engineers the two-layer recurrent block as an induction-head circuit (layer-1 attention copies variable identities to the RHS; layer-2 attention retrieves their values via induction; the MLP performs modular addition in the frequency domain, verified via 3D DFT). The error attribution (Table 2: 40 errors in 23k examples, dominated by feedforward arithmetic) and the ⟨EMPTY⟩-discounted-loss variant removing oracle depth supervision (Appendix C) are welcome additions.

**Strengths.** The task design is clean and well-motivated for controlled study. The mechanistic analysis is unusually thorough and, for the specific 2-layer probed model, largely convincing—the relative-variance probing, norm-amplification decomposition, and DFT analysis are careful and internally consistent. The self-correction finding (that the model does *not* self-correct without corruption training, Appendix D.4.2) is honest and non-obvious. The ⟨EMPTY⟩ variant partially addresses the most obvious "supervision-is-too-strong" objection.

**Weaknesses (elaborated below).** The central empirical claims rest on single best-of-search runs with no reported seeds or error bars; the four mechanisms are never cleanly factorially ablated; the interpretability analysis is conducted on a *different* model (2-layer, no self-correction) than the headline model (4-layer ⟲), creating a claim/evidence mismatch; several theoretical framing statements are loose; and the single-task scope, while defensible, is asserted to generalize far beyond what is shown. The paper also contains multiple unretracted internal editing notes ("Add corresponding graph for this", "May need to update fig xaxis... – Awni"), which are integrity/quality concerns I flag explicitly.

**Audience:**

Yes

**Audience Explanation:**

This is unambiguous. Length/depth generalization and latent-space reasoning are highly active topics, and this paper's combination of (a) a clean, complexity-parameterized testbed, (b) a concrete recipe combining recurrence + intermediate latent supervision + discretization, and (c) an unusually detailed mechanistic reverse-engineering (induction head + Fourier modular addition, with quantitative error attribution) will be of clear interest to researchers working on algorithmic reasoning, mechanistic interpretability, recurrent/looped Transformers, and inductive biases for compositional generalization. The mechanistic section in particular is a genuine contribution: it is rare to see a proposed architecture accompanied by this level of circuit-level explanation and honest negative findings (e.g., the no-self-correction-without-corruption result). Even a reader skeptical of the generality claims would find the testbed and the interpretability methodology useful. The audience criterion is comfortably met.

**Broader Impact Concerns:**

No specific broader-impact concerns require a Broader Impact Statement for this work in its current form. The paper studies algorithmic generalization on a synthetic modular-arithmetic task using small Transformers trained from scratch; it does not involve human-subjects data, personal data, generative content with misuse potential, or dual-use capabilities beyond generic advances in reasoning architectures. The work is foundational and methodological.

I would only note: as a non-blocking observation that the authors themselves motivate the work by its relevance to scaling reasoning in large language models (§6.5). If the mechanisms were later integrated into large-scale systems, the usual considerations around more capable automated reasoning would apply, but this is speculative and well outside the scope of what is demonstrated here. A one- to two-sentence statement acknowledging that the techniques are intended as general inductive biases for reasoning systems would be sufficient if the editors wish to include one, but I do not consider its absence a deficiency.

**Claims And Evidence:**

No

**Claims Explanation:**

Let me be precise about *which* claims fail, because the paper is a mixture of well-supported and poorly-supported assertions, and TMLR's bar is exactly this question.

**1. The headline generalization claim lacks statistical support.** The core empirical claim "near-perfect performance across all OOD splits" is carried entirely by Figure 3a/13 and Figure 11. Nowhere in the paper are there error bars, confidence intervals, standard deviations, or a reported number of seeds for the main comparison. Appendix D states the interpretability model was chosen by "cherry-pick[ing] the best-performing model trained with the same configuration with different random seeds" (99.98% Fully Solved). This tells me seeds *were* run, but only the maximum is reported, and the *variance* is never shown. For a paper whose entire thesis is "these mechanisms robustly enable generalization," the run-to-run robustness is exactly the quantity that must be reported. A best-of-N maximum over an unstated N is not evidence of robustness it is a selection artifact. This alone is disqualifying under criterion 1 until fixed, and it is straightforwardly fixable.

**2. The four mechanisms are not cleanly ablated, so the causal attribution to each mechanism is unsupported.** Table 1 defines the method ladder, but it confounds multiple axes. Consider what is actually varied:

- Continuous Latent Space Supervision = M1 + M2 (partial), no M3, no M4.

- Discrete Latent Space Supervision = M1 + M2 + M3, no M4.

- Discrete Latent Space Supervision ⟲ = M1 + M2 + M3 + M4.

This is a *nested chain*, not a factorial ablation. Critically, the ⟲ model uses **4 layers (D384)** while the non-⟲ discrete model uses **2 layers (D256)** (Appendix B.5, D). So the jump from "Discrete" to "Discrete ⟲" confounds *self-correction* with *doubling the recurrent-block depth and widening the model*. The paper even acknowledges error correction "requires more layers" (§4.2), but that means the Figure 3a improvement attributed to Mechanism 4 is inseparable from an increase in per-step compute and parameters. To claim Mechanism 4 helps, you need the 4-layer discrete model *without* self-correction as the control. Figure 11 lists "DeBERTa-L4H16D256" (no ⟲) at ~0.97 and "DeBERTa-L4H16D384 ⟲" at ~1.0, so the marginal contribution of self-correction, holding depth fixed at 4 layers, appears *small*, directly undercutting the prominence given to Mechanism 4 in the narrative. This needs to be stated honestly.

Similarly, there is no ablation isolating **algorithmic supervision (M2) from discretization (M3)** independent of recurrence, nor a "discretization without supervision" cell. The claim that each mechanism contributes is therefore only partially supported.

**3. The interpretability analysis is performed on a model that is not the headline model.** This is the most serious internal inconsistency. Section 5 and all of Appendix D analyze **DeBERTa-L2H16D256, Discrete Latent Space Supervision *without* the re-correction mechanism**. But the paper's central artifact, the one achieving "near-perfect" generalization and appearing first in every legend, is the 4-layer ⟲ model. So the mechanistic story ("here is the induction-head + modular-addition circuit that explains robust OOD generalization") explains a *weaker* model, and the paper provides **no** mechanistic evidence for what the self-correction dynamics actually do inside the headline model. The abstract promises "mechanistic interpretability analysis showing how these mechanisms yield robust OOD generalization" (plural, all mechanisms). In fact, §D.4.2 shows the probed 2-layer model provably does *not* self-correct. So the interpretability section characterizes an architecture missing the very mechanism the paper foregrounds. The claim, as written, overreaches its evidence.

**4. The "perfect generalization" phrasing versus the actual curves.** Figure 12d (the honest, zoomed-in plot) shows the best models *declining* with N: L4H16D256 and L4H16D384 drift from ~1.0 toward ~0.75–0.95 by N=128, and L4H16D256 (green) falls to ~0.68. This is good performance, but it is not "perfect," and it is visibly *monotonically degrading*, consistent with slow error accumulation, rather than a truly depth-invariant solution. The main-text Figure 3a, plotted on a full 0–1 axis, obscures this decay. The abstract's "perfect generalization on inputs several times larger" and §4.2's "near-perfect performance across all OOD splits" should be softened to match Figure 12d, or the discrepancy should be explained. Claiming depth-invariance while your own fine-grained plot shows depth-dependent decay is a claim/evidence mismatch.

**5. Loose theoretical framing.** The "Theoretical view" box (§3.2) argues fixed-depth Transformers can't solve the task because circuit evaluation is P-complete, and constant-depth logarithmic-precision Transformers are in TC⁰ ⊊ P (conjectured). This is fine as motivation but is stated with more force than warranted:

- The P-completeness of the Circuit Value Problem [Lad75] concerns *Boolean* circuit evaluation; the reduction to *mod-p* arithmetic circuits is asserted ("logical gates can be expressed as arithmetic operations mod p=2"), but the task actually studied uses **p=23** and a *bounded* N≤128. Asymptotic circuit-complexity separations say nothing rigorous about a finite, small-N regime. The argument establishes at most a heuristic motivation, not a theorem about this task, and should be labeled as such.

- "no polylog-depth parallel algorithm exists... unless P=NC" correct in spirit, but the relevance to a recurrent Transformer whose depth T scales *linearly* with circuit depth is that the model is precisely *not* polylog-depth. This should be made explicit; as written, it reads as if the theory supports the proposed method, when it only rules out a strawman (fixed constant depth).

**6. Baseline fairness caveat.** The paper reports best-of-search for all methods, which is good. But CoT is fundamentally disadvantaged on this task in a way worth stating: the CoT trace length grows super-linearly with N (every node's full equation + value), so at N=128 the sequences are extremely long, and the failure may partly reflect a length-extrapolation problem in the *positional encoding over CoT tokens* rather than an intrinsic limitation of CoT-style supervision. The comparison "our latent recurrence beats CoT" partly reduces to "short latent rollouts extrapolate better than very long token rollouts," which is informative but should be framed carefully rather than as evidence that latent reasoning is categorically superior.

The paper is well above the threshold of *interesting and largely honest work*, but as written, several of its headline causal and quantitative claims are not supported to TMLR's standard. Most issues are fixable in revision, which is why my requested changes distinguish critical from strengthening.

**Requested Changes:**

I group these by whether they are **[CRITICAL]** (required to move me to acceptance) or **[STRENGTHENING]** (would improve the paper but are not blocking).

**[CRITICAL] 1. Report seeds and variance for all main results.** For Figure 3a/13, Figure 11, and Figure 12, report the number of random seeds and show mean ± std (or min/max/median, or box plots) rather than a single best run. The paper already ran multiple seeds (Appendix D admits cherry-picking the best); please report the distribution. If run-to-run variance is high, that must be disclosed and discussed. Without this, the robustness claim central to the paper is unsupported.

**[CRITICAL] 2. Provide a clean factorial ablation, and in particular disentangle self-correction (M4) from recurrent-block depth.** At minimum, add the **4-layer Discrete Latent Space Supervision *without* self-correction** as a control against the 4-layer ⟲ model, so the marginal effect of M4 at fixed depth/width is visible. Figure 11 hints this marginal effect is small; the main text should reflect whatever the data show. Ideally, present a factorial table over {M2 on/off} × {M3 on/off} × {M4 on/off} at matched depth/width, with seeds. If a full factorial is infeasible, justify the specific cells chosen and do not attribute gains to a mechanism whose effect is confounded with capacity.

**[CRITICAL] 3. Resolve the interpretability/headline-model mismatch.** Either (a) redo (or add) the mechanistic analysis on the 4-layer ⟲ model that is the paper's headline result, especially characterizing what the self-correction dynamics do internally; or (b) explicitly and prominently reframe the interpretability claims to state that they characterize the 2-layer *non*-self-correcting model, and correspondingly soften the abstract/intro ("mechanistic analysis showing how *these mechanisms* yield robust OOD generalization" → "how *recurrence, supervision, and discretization* yield…"). As written, the abstract claims mechanistic support for all four mechanisms, while §D.4.2 shows the probed model does not even perform the behavior Mechanism 4 targets. This inconsistency must be removed.

**[CRITICAL] 4. Align quantitative language with Figure 12d.** The best models *decline* with N (down to ~0.68–0.95 at N=128 depending on config). Replace "perfect generalization" / "near-perfect performance across all OOD splits" with language consistent with the observed monotonic decay, or explain why the full-axis Figure 3a is the fairer summary than the zoomed Figure 12d. Please also report the exact "% Fully Solved" numbers at N=128 for the headline model(s) in a table, not only in plots.

**[CRITICAL] 5. Remove unretracted internal notes and finalize the manuscript.** Delete the "Add corresponding graph for this" placeholder (App. A.2.1) and the "May need to update fig xaxis... – Awni" margin note (App. D.4), and add the referenced CoT example graph if it was intended. These indicate an unfinished submission.

**[CRITICAL] 6. Correctly scope the complexity-theoretic argument.** State clearly that the TC⁰ ⊊ P and P-completeness arguments (i) concern asymptotic worst-case Boolean circuit evaluation, (ii) are conjectural (TC⁰ ⊊ P is not proven), and (iii) do not constitute a theorem about the finite N≤128, p=23 regime actually studied. Reframe as motivation, not as a result. Explicitly connect the argument to the fact that your recurrence uses depth T linear in circuit depth (i.e., you deliberately escape the constant-depth regime).

**[STRENGTHENING] 7. At least one additional task/primitive set.** The paper's §6.3–6.4 argue extensively that the four mechanisms are general (any program = DAG). This is asserted, not shown. A single additional instantiation—e.g., a different operation set, Boolean circuits, or a small dynamic-programming task—would move these from claims to evidence. I accept the single-task scope as a legitimate controlled study for TMLR and am not blocking on this, but the generality prose should be toned to match the evidence, or backed by one more task.

**[STRENGTHENING] 8. Fairer CoT framing / a length-matched analysis.** Discuss the confound that CoT traces grow super-linearly in N and that CoT failure may partly reflect positional-encoding length-extrapolation over very long token sequences rather than an intrinsic CoT limitation. A useful control: report CoT token-sequence lengths at N=32 vs N=128, and if possible a CoT variant with the best available long-context positional scheme.

**[STRENGTHENING] 9. Halting-rule robustness.** The fixed-point halting rule (iterate until z^(t+1)=z^(t)) is elegant, but the paper does not report how often the fixed point is (a) reached, (b) reached at the *correct* answer vs. a wrong attractor, or (c) never reached (oscillation/non-convergence). Please quantify failure modes of the halting criterion, and report the actual distribution of iteration counts vs. circuit depth (Figure 3b/14 show solve-rate vs. iterations but not the halting behavior per se).

**[STRENGTHENING] 10. Details on the self-correction perturbation.** Specify the corruption probability, which factors are corrupted, and sensitivity to these choices. The mechanism's effect should be shown to be robust to the perturbation hyperparameters, not tuned to a single lucky setting.

**[STRENGTHENING] 11. ⟨EMPTY⟩-discounted-loss variant (Appendix C) is under-reported.** This is one of the more important results for the paper's generality argument (removing oracle depth), yet it is "preliminary" with only Figure 13/14 and no seeds, no ablation of α, and no quantitative comparison table. Promoting this to a properly-evaluated result (with α sensitivity and seeds) would substantially strengthen the paper's claim that intermediate supervision is not strictly required.

**[STRENGTHENING] 12. Minor clarity items.** (a) Equation (3) in the main text and Eq. (5)–(6) in Appendix B appear to sum only the value-factor loss in Eq. (6) despite the text saying "sum of the individual factor losses"—please fix the indexing so the summed factors are explicit. (b) Define "Depth(x_i)" for leaf nodes explicitly (depth 1 per §D.4.1, but the main text is ambiguous). (c) Figure 5 (right) and Figure 16 use "others"/"unimportant" inconsistently for the same head group—unify the terminology.

---

### Review · Reviewer_4HPa · 2026-07-09

**Summary Of Contributions:**

The paper studies OOD (length/depth) generalization in Transformers on a controlled testbed: evaluating modular-arithmetic circuits (DAGs, mod 23), trained on graphs with ≤32 nodes and tested up to 128. It proposes four mechanisms — (1) input-adaptive recurrence, (2) depth-indexed algorithmic supervision in latent space, (3) a discrete latent bottleneck that "anchors" representations across iterations, and (4) a self-correction scheme trained by corrupting latent states. Combined, the full model reaches near-perfect "fully solved" accuracy well beyond the training regime, where CoT and end-to-end baselines degrade around N≈40. The paper adds a detailed mechanistic analysis showing a two-step induction-head circuit (copy variable name → retrieve value) plus a Fourier-domain modular-addition MLP, and an error attribution (40 errors in 23k, mostly in the final MLP). Appendix C sketches a variant that removes the oracle depth labels.

**Additional Comments:**

The paper is genuinely nice work and I'd like to see it in TMLR. My concerns are about matching claims to evidence and finishing the manuscript, not about the core idea. One flag for the AE: the "– Awni" notes are an anonymity leak and should be handled before the paper goes further.

**Audience:**

Yes

**Audience Explanation:**

The task design is clean, the OOD result is strong, and the mechanistic analysis is careful. Readers working on length/depth generalization, latent/recurrent reasoning, and mechanistic interpretability would find it relevant and useful.

**Claims And Evidence:**

Yes

**Claims Explanation:**

The claims are partially supported. The central empirical claim that the combined mechanisms give sharp OOD generalization on this task is convincing and clearly presented. However, several claims currently outrun the evidence:

- Mechanism 4 is not backed by the interpretability analysis. The mechanistic study is done on the discrete model without self-correction, and Appendix D.4.2 explicitly states that this model does not self-correct. So the "why it's robust" story never actually validates the error-correction mechanism.
- Generality claims are broad but evidence is one task. Sections 6.3–6.4 talk about a "general architectural template" for "generic DAG-structured computations," but everything rests on a single mod-23 arithmetic task.
- No variance is reported. The main comparison (Fig. 3a) reports the single best model per method, and the interpretability model is a cherry-picked best seed (99.98%). Without seed variance it's hard to judge robustness of the comparison.
- The "weaker supervision than CoT" claim is not quantified. The method supervises every node's value at every depth and teacher-forces the discrete states across iterations (App. B.3), which is not obviously weaker than a CoT trace.
- Appendix C (no depth oracle) is the key experiment for the "supervision isn't too strong" question, but it is only qualitative — no numbers or tables.

**Requested Changes:**

Major changes:

- Remove leftover author notes and finish the draft. There are TODO comments signed "– Awni" in the text (p.22 and p.43). These break double-blind and show the draft is incomplete — please clean these up and complete the missing figures/examples.
- Fix the Mechanism-4 evidence gap. Either add a mechanistic analysis of the model with self-correction (showing the attractor/error-correcting dynamics), or scope the interpretability claims to M1–M3 and stop implying the analysis explains error correction.
- Report variance. Give mean±std (or scatter) over several seeds for the main comparison, and state whether the "best model" selection was applied symmetrically to all methods.
- Match the generality claims to the evidence. Either add a second, structurally different task (e.g. Boolean-circuit or DP-style evaluation), or clearly downgrade Sections 6.3–6.4 to hypotheses/future work.
- Add quantitative results to Appendix C.

Please report full training details for reproducibility. Key hyperparameters are missing: optimizer, learning rate, batch size, number of training steps, and training-set size. Some method-specific settings are also unspecified (the corruption probability in M4, the weight-decay value, and the number of seeds used). A single hyperparameter table, ideally with a code release, would fix this.

Minor changes:

- Clarify the inference protocol: how is the number of iterations T(X) chosen at test time — fixed-point halting vs. using the true depth?
- Clarify the discretization training (straight-through vs. teacher forcing) and quantify supervision strength vs. CoT.
- Add an adaptive-halting baseline and/or a looped-Transformer baseline on this task.
- Note honestly that the dense intermediate supervision the method needs is hard to obtain in natural-language settings like GSM8K.
- Soften wording like "perfect"/"arbitrary lengths" to the tested range (train ≤32, test up to 128, ~4×); make the paragraph clearly motivational.

---

### Review · Reviewer_dLLj · 2026-07-19

**Summary Of Contributions:**

This paper studies out-of-distribution generalization in Transformers on a controlled modular arithmetic circuits task, with a focus on length/depth extrapolation beyond the training regime. The main idea is to replace token-level chain-of-thought supervision with latent-space reasoning implemented through four mechanisms: input-adaptive recurrence, algorithmic supervision on latent states, discretized/anchored latent representations, and an explicit self-correction scheme. Empirically, the full method substantially improves OOD generalization over end-to-end and CoT baselines, and appears to achieve near-perfect performance on much larger circuits than seen during training. The paper also includes mechanistic interpretability analysis suggesting that the learned solution involves variable-copying attention patterns, induction-like retrieval of values, and modular addition in the final feedforward layer.

The paper is interesting because it moves beyond simply showing that recurrence or CoT helps, and instead proposes a coherent architectural recipe for scalable latent-space algorithm execution. The strongest aspects are the clear problem setup, the systematic decomposition into four mechanisms, the convincing performance trend across ablations, and the effort to connect behavior to internal circuits. The main weakness is that the evidence is confined to a synthetic domain, so the broader claim of “unlocking” OOD generalization in Transformers should be interpreted more narrowly. In addition, several mechanisms are introduced together, and while the ablations are helpful, it remains somewhat unclear how broadly these findings transfer beyond this task family.

**Audience:**

Yes

**Audience Explanation:**

Yes. I expect this paper to be of interest to readers working on reasoning, out-of-distribution generalization, recurrent architectures, algorithmic learning, and mechanistic interpretability. The paper addresses a central question in modern ML: what architectural biases help models learn scalable procedures rather than shallow pattern matching.

Even though the experimental domain is synthetic, the paper provides a clean and well-controlled setting for studying a difficult problem that is often hard to isolate in natural-language benchmarks. The architectural ideas and the interpretability findings are likely to be useful to researchers thinking about latent reasoning, recurrent inference, and test-time scaling.

**Broader Impact Concerns:**

I do not see major immediate ethical concerns specific to this work. The paper studies reasoning and generalization in a synthetic environment, and its main contribution is methodological rather than application-specific. If the ideas transfer, the likely impact is mostly positive for building more robust and interpretable reasoning systems.

A minor concern is one of scientific communication rather than societal harm: broad claims about “unlocking” OOD generalization may be overread if not carefully scoped. It would therefore be beneficial for the final version to clearly state the limits of the current evidence and avoid implying broad real-world reasoning capabilities beyond what is demonstrated.

**Claims And Evidence:**

Yes

**Claims Explanation:**

Within the scope of the modular arithmetic circuits task, the empirical evidence is generally strong. The paper compares standard end-to-end and CoT baselines against progressively richer variants of the proposed architecture, and the trends are consistent: recurrence helps, latent algorithmic supervision helps more, discretization is especially important for stable scaling, and self-correction further improves robustness. The test-time scaling behavior also matches the paper’s central claim that adaptive recurrent computation can support larger inputs beyond the training regime.

The mechanistic analysis is a meaningful strength because it does more than visualize attention; it attempts to identify concrete functional roles for components, such as copying variable identities, retrieving prior values through induction-like attention, and performing modular addition in the MLP. That said, some of the broader framing should be softened. The evidence strongly supports the paper’s conclusions on this synthetic task, but it is less clear that the same mechanisms will unlock robust OOD generalization in more realistic reasoning domains or in large language models more broadly.

**Requested Changes:**

Major

1. Narrow and calibrate the high-level claims. The title and framing currently suggest a broad result about unlocking OOD generalization in Transformers, but the evidence is limited to a synthetic modular arithmetic circuits setting. The paper would be stronger if it more explicitly positioned its conclusions as a compelling demonstration in a controlled domain rather than a general solution for Transformer reasoning.

2. Clarify the role of additional test-time compute versus the role of the proposed architectural biases. Since the method relies on input-adaptive recurrence, part of the gain may come from scaling compute at inference time. The paper should more clearly distinguish improvements due to “more steps” from improvements due to discretization, algorithmic supervision, and self-correction.

3. The paper should more directly discuss what properties of the modular arithmetic circuits task are expected to transfer to broader reasoning problems, and which conclusions may be task-specific.


Minor

1. Improve exposition around the discrete bottleneck and factorization of latent states. This component seems central to the success of the method, and a slightly more concrete explanation would make the contribution more accessible.